# Selective deuteration as a tool for resolving autoxidation mechanisms in $\alpha$-pinene ozonolysis

Melissa Meder[1], Otso Peräkylä[1], Jonathan G. Varelas[2], Jingyi Luo[2], Runlong Cai[1], Yanjun Zhang[1,4], Theo Kurtén[1,3], Matthieu Riva[4], Matti P. Rissanen[5,3], Franz M. Geiger[2], Regan J. Thomson[2], and Mikael Ehn[1]

[1]Institute for Atmospheric and Earth System Research (INAR/physics), University of Helsinki, Helsinki, Finland
[2]Department of Chemistry, Northwestern University, Illinois, USA
[3]Department of Chemistry, University of Helsinki, Helsinki, Finland
[4]Univ Lyon, Université Claude Bernard Lyon 1, CNRS, IRCELYON, Villeurbanne, France
[5]Aerosol Physics laboratory, Tampere University, Tampere, Finland

**Correspondence:** Melissa Meder (Melissa.Meder@helsinki.fi) and Mikael Ehn (Mikael.Ehn@helsinki.fi)

**Abstract.** Highly oxygenated organic molecules (HOMs) from $\alpha$-pinene ozonolysis have been shown to be significant contributors to secondary organic aerosol (SOA), yet our mechanistic understanding of how the peroxy radical-driven autoxidation leads to their formation in this system is still limited. The involved isomerisation reactions such as H-atom abstractions followed by $O_2$ additions can take place on sub-second time-scales in short-lived intermediates, making the process challenging

to study. Similarly, while the end products and sometimes radical intermediates can be observed using mass spectrometry, their structures remain elusive. Therefore, we propose a method utilising selective deuterations for unveiling the mechanisms of autoxidation, where the HOM products can be used to infer which C-atoms have taken part in the isomerisation reactions. This relies on the fact that if a $C-D$ bond is broken due to an abstraction by a peroxy group forming a $-OOD$ hydroperoxide, the D-atom will become labile and able to be exchanged with a hydrogen atom in water vapour ($H_2O$), effectively leading to loss

of the D-atom from the molecule.

In this study, we test the applicability of this method using three differently deuterated versions of $\alpha$-pinene with the newly developed chemical ionisation Orbitrap (CI-Orbitrap) mass spectrometer to inspect the oxidation products. The high mass resolving power of the Orbitrap is critical, as it allows the unambiguous separation of molecules with a D-atom ($m_D$=2.0141) from those with two H-atoms ($m_{H2}$=2.0157). We found that the method worked well and we could deduce that two of the three

tested compounds had lost D-atoms during oxidation, suggesting that those deuterated positions were actively involved in the autoxidation process. Surprisingly, the deuterations were not observed to decrease HOM molar yields, as would have been expected due to kinetic isotope effects. This may be an indication that the relevant H (or D) abstractions were fast enough that no competing pathways were of relevance despite slower abstraction rates of the D-atom. We show that selective deuteration can be a very useful method for studying autoxidation on a molecular level, and likely not limited to the system of $\alpha$-pinene

ozonolysis tested here.

# 1 Introduction

Highly oxygenated organic molecules (HOMs) are atmospheric compounds with low to extremely low volatilities formed from volatile organic compounds (VOCs) in the atmosphere via autoxidation (Crounse et al., 2013; Ehn et al., 2014; Bianchi et al., 2019). HOMs formed from $\alpha$-pinene ozonolysis have been shown to contribute significantly to secondary organic aerosol (SOA) (Ehn et al., 2014), however, a mechanistic understanding of this system is limited, despite it having been studied computationally and experimentally (Kurtén et al., 2015; Iyer et al., 2021). Autoxidation involving peroxy radicals ($RO_2$) have been shown to be the source of the high oxygen content (Berndt et al., 2018), but the exact route or routes of isomerisation cannot be derived purely by the mass spectrometric methods (Jokinen et al., 2012; Riva et al., 2019a) typically used to detect the HOMs. In addition, the isomerisation reactions are very fast (Bianchi et al., 2019) which makes following them in real-time typically not possible. One way to circumvent these limitations is to perturb the autoxidation process in a way that leads to different HOM products depending on the reaction mechanism.

Deuteration is a commonly used method for multiple applications in chemistry, including atmospheric studies (Rissanen et al., 2014; Zhang et al., 2017; Ye et al., 2018; Michelotti and Roche, 2019; Wang et al., 2020; Li et al., 2021). In deuteration, one or more hydrogen atoms in a molecule are exchanged to deuterium atoms (D). Deuteration can be a full deuteration where all hydrogen atoms are exchanged to deuterium atoms as used by Rissanen et al. (2014) or it can be done selectively, where specific hydrogen atoms are exchanged with deuterium atoms as used by Zhang et al. (2017) and Ye et al. (2018). The deuteration can have several impacts on the reactions taking place, including a decreased rate of reaction due to the higher mass of D-atom compared to H-atom, the so-called kinetic isotope effect (Laidler, 1987). Rissanen et al. (2014) used fully deuterated cyclohexene and found that the ozonolysis formed HOMs at greatly reduced yields compared to the non-deuterated cyclohexene. They also showed that D-atoms were exchanged to H-atoms in contact with water vapour in cases where the C$-$D bond was broken, and the D became attached to an oxygen through an O-D bond. In other words, the autoxidation was perturbed in predictable ways by the deuteration. Nevertheless, we expect that selective deuteration can provide substantially more information still. To our knowledge, selective deuteration has not been utilised in studying autoxidation pathways from $\alpha$-pinene ozonolysis before.

In this study, we inspected $\alpha$-pinene ozonolysis by simulating atmospheric conditions in a reaction chamber. We used non-deuterated as well as three selectively deuterated precursors to test how the resulting HOM spectrum might change, and to gain insight into the autoxidation processes. We used a chemical ionisation Orbitrap mass spectrometer (Riva et al., 2019a) which has a high enough resolving power to unambiguously distinguish between the different isotopes in the spectra. We compared the spectral distribution of different HOMs as well as the overall HOM yields for the different precursors. In particular, we examined the numbers of D-atoms lost for the differently deuterated precursors for different HOM molecules, with the aim to assess from which C-atoms H/D-abstractions take place during autoxidation. We also compare our findings to the few suggested mechanisms from earlier publications.

## 2 Selective deuteration and autoxidation

In this section, we first describe the selectively deuterated precursors used in this study (Sect. 2.1), after which we describe the main reaction pathways during $\alpha$-pinene ozonolysis that are of relevance for this study (Sect. 2.2). Lastly, we describe how selective deuteration can be used to study the oxidation processes (Sect. 2.3).

### 2.1 Selectively deuterated precursors

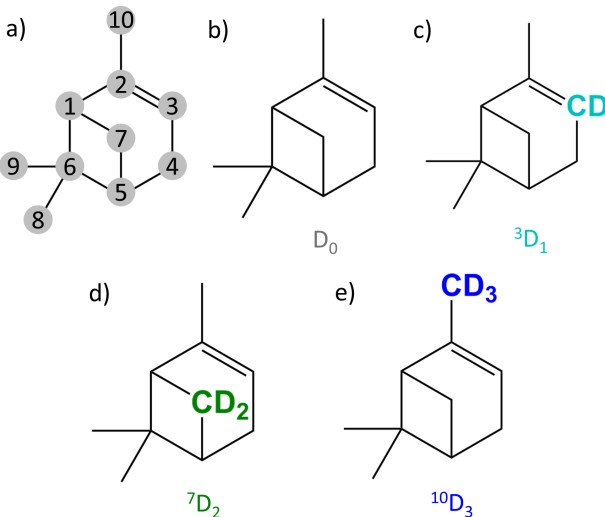

**Figure 1.** $\alpha$-pinene with numbered carbons following IUPAC numbering (a) and the molecular structures of the selectively deuterated precursors used in this study (b-e). The precursors are named based on the number of the carbon that is deuterated and the number of deuterium atoms added to the molecule. The carbons with the deuterium atoms are highlighted with colours specific to the precursors. b) Normal $\alpha$-pinene is $D_0$, c) $\alpha$-pinene with 1 deuterium is $^3D_1$ (cyan), d) $\alpha$-pinene with 2 deuterium atoms is $^7D_2$ (green), and e) $\alpha$-pinene with 3 deuterium atoms is $^{10}D_3$ (blue).

In this study, we used four precursors and their chemical structures are shown in Fig. 1. The three selectively deuterated precursors were synthesised in Northwestern University, Illinois, USA. The precursors are named based on the carbon that was deuterated and the number of deuterium atoms they have: $D_0$ is standard $\alpha$-pinene molecule without any isotopic alteration ($C_{10}H_{16}$), $^3D_1$ is $\alpha$-pinene with one deuterium atom on C-3 ($C_{10}H_{15}D$, $\alpha$-pinene-3-*d1*; see Fig. 1), $^7D_2$ is $\alpha$-pinene with two deuterium atoms on C-7 ($C_{10}H_{14}D_2$, $\alpha$-pinene-7,7-*d2*), and $^{10}D_3$ is $\alpha$-pinene with three deuterium atoms on C-10 ($C_{10}H_{13}D_3$, $\alpha$-pinene-10,10,10-*d3*). The vinylic C-3 carbon and the allylic C-10 methyl group carbon are expected to undergo changes during the ozonolysis process (Jenkin et al., 1997; Kurtén et al., 2015), while the cyclobutyl ring C-7 carbon has not to our knowledge been shown to take part in any reactions in the initial ozonolysis reaction.

Synthesis of the $^{10}D_3$ and $^7D_2$ $\alpha$-pinene species was carried out according to the published literature procedures (Upshur et al., 2016, 2019) while access to the $^3D_1$ $\alpha$-pinene was accomplished through a new three-step route utilising nopinone as the starting material (see Supporting Information for complete synthetic details). The purity of each compound was >95 % as determined by $^1$HNMR spectroscopy and we were unable to observe residual proton resonances associated with the deuterated carbon positions for any sample, indicating complete deuterium incorporation within the detection limits of the spectrometer (see Supporting Information for $^1$HNMR spectral data).

## 2.2  Autoxidation in $\alpha$-pinene ozonolysis

As discussed above, autoxidation is a major source of condensable low-volatile vapours from $\alpha$-pinene ozonolysis. The reaction of $\alpha$-pinene with ozone starts by the attachment of ozone to the double bond, followed by bond scission and rapid isomerisations via a Criegee intermediate and vinyl hydroperoxide, and OH loss, potential isomerisation and $O_2$ addition to form a primary peroxy radical ($C_{10}H_{15}O_4$), as shown in Fig. 2a (Kurtén et al., 2015; Iyer et al., 2021). From these peroxy radicals, the process can continue with autoxidation, which is successive H-shifts and oxygen additions. For pathways I to III, Kurtén et al. (2015) calculated that subsequent H-shifts starting from these primary $RO_2$ were too slow to explain observed HOM formation rates. This was primarily due to the intact cyclobutyl ring causing strain for the abstraction of hydrogen atoms across the ring. However, Iyer et al. (2021) found that an isomerisation reaction was available for the vinoxy intermediate in pathway I due to excess energy remaining from the initial ozonolysis reaction. The excess energy allows the breaking of the cyclobutyl ring (pathway IV) and subsequent rapid H-shifts leading to $RO_2$ with up to eight oxygen atoms (Fig. 2b). The $O_8-RO_2$ product remains the only experimentally and computationally supported HOM-forming pathway in this system (Rissanen et al., 2014; Iyer et al., 2021), and is thus the most promising candidate to explain many of the observed closed-shell HOMs following termination reactions.

The autoxidation process can be terminated via unimolecular or bimolecular reactions, and the most relevant ones are introduced below. A common unimolecular reaction (R1) of an $RO_2$ that has undergone autoxidation is the H-abstraction from a C-atom with a hydroperoxide functionality. This rapidly leads to the loss of a hydroxyl radical and the formation of a closed shell HOM with a carbonyl functionality at said C-atom (Crounse et al., 2013). The $RO_2$ can also undergo different bimolecular termination reactions (Atkinson and Arey, 2003). In the absence of $NO_x$, as in this study, there are two relevant reaction partners, namely $HO_2$ and other $RO_2$, and their main reaction products are listed below (R2-R6). In brief, $RO_2$ cross reactions can form closed shell monomers (R2), ROOR accretion products which are commonly referred to as dimers (R3), or alkoxy radicals RO (R4), with the latter able to undergo further isomerization or termination reactions. Reactions of $RO_2$ with $HO_2$ typically form hydroperoxides ROOH (R5), but can also form RO (R6), especially for complex enough $RO_2$ (Hasson et al., 2005; Orlando and Tyndall, 2012; Iyer et al., 2018).

$$HOORO_2\cdot \rightarrow O{=}R_{-H}OOH + OH\cdot \qquad (R1)$$

$$RO_2 \cdot + R'O_2 \cdot \rightarrow ROH + R'_{-H}{=}O + O_2 \tag{R2}$$

$$RO_2 \cdot + R'O_2 \cdot \rightarrow ROOR' + O_2 \tag{R3}$$

$$RO_2 \cdot + R'O_2 \cdot \rightarrow RO \cdot + R'O \cdot + O_2 \tag{R4}$$

$$RO_2 \cdot + HO_2 \cdot \rightarrow ROOH + O_2 \tag{R5}$$

$$RO_2 \cdot + HO_2 \cdot \rightarrow RO \cdot + OH \cdot + O_2 \tag{R6}$$

## 2.3 Studying autoxidation with selective deuteration

During the oxidation and autoxidation processes, hydrogen atoms can be lost from the precursor molecule in multiple ways. If a carbon loses a hydrogen atom during a H-shift, the carbon has actively participated in the autoxidation process and we refer to this as the carbon being "active". More specifically, when discussing the experimental results, we only consider the carbon to be active when we observe the D-abstractions take place at fractions above our uncertainty margins. While we are primarily interested in H-shifts by $RO_2$, they can also occur by RO radicals or in the Criegee intermediate step, a reaction that takes place in all cases.

When a hydrogen atom is shifted away from a carbon, the hydrogen atom ends up in a hydroperoxide group $-OOH$, except in the case of RO H-shifts, where the result is a hydroxyl group $-OH$. In both cases, the H-atoms are now labile enough that they can be exchanged in collisions with water vapour molecules. In the special case where a deuterated carbon has been active, and a D-shift has taken place, the D can eventually be lost from the molecule entirely (example shown in Fig. 2c). With deuteration, we can thus observe the change in the molecular mass when the D/H exchange has taken place, and thus specify which carbon in the precursor has been active. Additionally, if a deuterated precursor produces a mass spectral signature for a given HOM molecule where part of the signal has lost a D while another has not, it is an indication of different isomers for this composition. In other words, several pathways lead to the same elemental composition, but with different C-atoms being active.

If the D-atom was internally abstracted by the Criegee intermediate, it will be lost by ejection of the OD, as would be the case in pathway II (Fig. 2a, blue H) for $^{10}D_3$ precursor. Furthermore, deuterium atoms can be lost if the selectively deuterated carbon itself is lost through fragmentation during the HOM formation reaction.

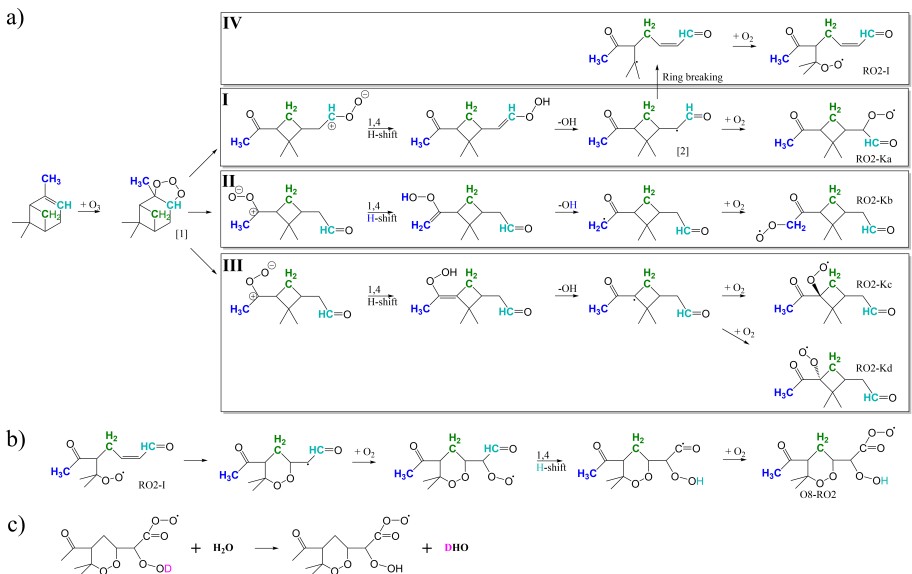

**Figure 2.** a) Reaction pathways for $\alpha$-pinene ozonolysis up until the primary peroxy radical ($RO_2$) from Kurtén et al. (2015) (I-III) and Iyer et al. (2021) (IV). b) Continued reaction pathway for the IV pathway up to a peroxy radical containing 8 oxygen atoms ($O_8-RO_2$) (Iyer et al., 2021). In a) and b) the carbon and hydrogen atoms corresponding to deuterated ones in the selectively deuterated precursors are coloured matching the colouring in Fig. 1, i.e. cyan corresponds to $^3D_1$ precursor, green corresponds to $^7D_2$ precursor, and blue corresponds to $^{10}D_3$ precursor. c) Deuterium atoms are readily exchanged between HOMs and water vapour if the deuterium is in an $-OD$ or $-OOD$ group, resulting in HOMs without deuterium and semiheavy water i.e. water molecule with one hydrogen and one deuterium instead of two hydrogen atoms (Rissanen et al., 2014).

Only two of the four pathways discussed earlier result in the precursor losing a deuterium atom. In the four pathways shown in Fig. 2a-b, the carbon and hydrogen atoms corresponding to those of the selectively deuterated precursors are highlighted with colours matching the colouring from Fig. 1. The atoms coloured with cyan correspond to $^3D_1$ precursor, the green ones correspond to $^7D_2$ precursor, and the blue ones correspond to $^{10}D_3$ precursor. In pathway II, the $^{10}D_3$ precursor would lose one deuterium when OH is fragmented from the molecule, and in pathway IV with the continued steps in Fig. 2b, the deuterium in $^3D_1$ precursor would be shifted to a hydroperoxy group and hence readily exchanged to a hydrogen with water.

In addition to the deuteration impacting the molecular mass observed in the mass spectra, the H-shifts themselves will become much slower since the deuterium atom being abstracted has twice the mass of a hydrogen atom. Rissanen et al. (2014) found that the HOM yields were roughly 100 times lower for ozonolysis of fully deuterated cyclohexene $C_6D_{10}$ compared to that of standard cyclohexene $C_6H_{10}$. Some of the same effects should play a part in this oxidation system, namely the kinetic isotope effect and the slowing down of quantum tunnelling. The kinetic isotope effect is caused by the lower vibrational frequencies for the $C-D$ bond compared to $C-H$ bond, and it decreases the rate of the abstraction (Laidler, 1987). Furthermore, quantum mechanical tunnelling is important for the H/D-abstractions, and this causes an even larger decrease to the reaction rate (Laidler, 1987).

The exact total decrease in the reaction rate coefficients depends on the radical structure, but a typical range is a factor of 20-100 slower for D-shifts versus H-shifts (Crounse et al., 2011; Praske et al., 2018). However, it is important to keep in mind that the effect on the likelihood of the D-shift taking place will still depend on the possible competing reactions. For example, if the H-shift is 1000 times faster than any competing reaction, substituting for a D-shift will barely impact the branching of the reaction, whereas with similar H-shift rates with any competing reaction, deuteration will largely shut down this pathway.

## 3   Methods

In this section, we first describe the experimental set-up and the used instruments (Sect. 3.1). Then we describe how the data was analysed (Sect. 3.2).

### 3.1   Experimental set-up

A schematic of the experimental set-up is shown in Fig. 3. The experiments were conducted in the $2 \ \mathrm{m}^3$ "COALA" chamber
(teflon FEP chamber, Vector Foiltec, Germany) (Riva et al., 2019b; Peräkylä et al., 2020), a Continuous Stirred-Tank Reactor (CSTR) at the University of Helsinki, Finland. We monitored relative humidity (RH) and temperature with a temperature and humidity probe (INTERCAP® HMP60, Vaisala). RH stayed below the detection limit of the probe (under 1 %) and temperature was 26.0 °C within ±1.5 °C range. We monitored $NO_x$ (NO and $NO_2$) concentrations with a NO-$NO_2$ analyzer (model T200UP, Teledyne) and the concentrations were under 0.3 ppb. We generated ozone by running 5 lpm clean air
through an ozone generator (Dasibi 1008-PC ozone generator). We ran the chamber at a stable $O_3$ concentration ($\sim 43 \pm 2$ ppb), monitoring the ozone concentration with a photometric $O_3$ analyser (model 400, Teledyne instruments; ozone monitor in Fig. 3). The total flow through COALA-chamber was kept at 50 lpm by adding 45 lpm flow of clean air generated in a zero air generator (AADCO, Series 737-14, Ohio, USA). As a result, we achieved an average residence time of $\tau_R \approx 40$ min inside the chamber. With these conditions, we estimate the $\alpha$-pinene loss rate $[\alpha\text{-pinene}]_{loss}$ to be 23 % using the equation

$$[\alpha\text{-pinene}]_{loss} = k_1[O_3], \tag{1}$$

where $k_1 = 9.06 \cdot 10^{-17} cm^3 s^{-1}$ is the reaction rate coefficient for $\alpha$-pinene ozonolysis at $T = 26$ °C (Atkinson et al., 2006).
    During an experiment, we aimed to keep the precursor concentration stable when injecting the precursor into the chamber. We ran the chamber as a "steady-state chamber", meaning we continuously fed precursor and $O_3$ into the chamber throughout the experiments, aiming for stable concentrations resulting from a balance between sources (inflow) and sinks (outflow, chemical
reactions, and wall losses). As such, the absolute precursor consumption varied with the amount of precursor that was fed, which often varied even across one experiment. We injected precursor into the chamber using one of two methods: an overflow set-up for small amounts of precursor, and a syringe pump set-up when there was enough precursor for using a 5 μl syringe to take a sample. In both cases, the evaporated $\alpha$-pinene was injected into the chamber with a small $25 - 150$ lpm $N_2$ flow. We only had enough of one deuterated precursor ($^3D_1$) to be injected with the syringe pump set-up. With the other deuterated
precursors ($^7D_2$ and $^{10}D_3$), there was barely any visible liquid in the glass bottle, thus we could not draw the precursors into

the syringe to use with the syringe pump. Nevertheless, flushing $N_2$ over the bottle did release enough precursor to provide a spectrum of oxidation products, though the overflow set-up was harder to control and produced a less stable concentration in the chamber than the syringe pump set-up. The $\alpha$-pinene injection set-ups are described in more detail in the appendix (Sect. A1).

We monitored the $\alpha$-pinene ozonolysis products with a $NO_3^-$ CI-Orbitrap. The Orbitrap (Q Exactive Plus Orbitrap, Thermo Scientific) has an ultrahigh mass resolution (280 000 Th/Th), meaning it is possible to separate between two hydrogen atoms ($m_{H2}$ =2.0157) and a deuterium atom ($m_D$ =2.0141) signal in the mass spectrum, hence enabling the use of selectively deuterated precursors. Coupled with the nitrate Chemical Ionisation inlet ($NO_3^-$ CI-inlet) which is selective towards highly oxidised products (Hyttinen et al., 2015), the instrument has been shown to be effective at detecting HOMs (Riva et al., 2019a;

Bianchi et al., 2019).

The precursor concentration was monitored using the proton transfer reaction time-of-flight (PTR-ToF) mass spectrometer (Ionicon Analytik GmbH, Austria; Yuan et al. (2017)). However, due to instrument failure, we conducted the last experiment with $^3D_1$ precursor using the Vocus-PTR-ToF (Tofwerk AG/Aerodyne Research, Inc.; Krechmer et al. (2018)) instead of the PTR-ToF. Nonetheless, both instruments are PTR-ToF mass spectrometers that can be used to monitor VOCs such as

monoterpenes (Yuan et al., 2017), and they were calibrated with the same method, hence they could fulfil the experimental purposes.

We used a 10 lpm sample flow rate with a 60 cm long sampling line for the CI-orbitrap, a 1 lpm sampling flow rate with a roughly 2.5 m sampling line for the ozone monitor, and a 5 lpm sampling flow rate with a roughly 1 m sampling line that was also split to house exhaust for the PTR-ToF mass spectrometers. As a result, we had a 16 lpm flow for sampling and the excess

flow from the chamber was fed to the house exhaust, keeping the chamber at a slight over-pressure to ensure contaminations from the room air could not leak into the chamber.

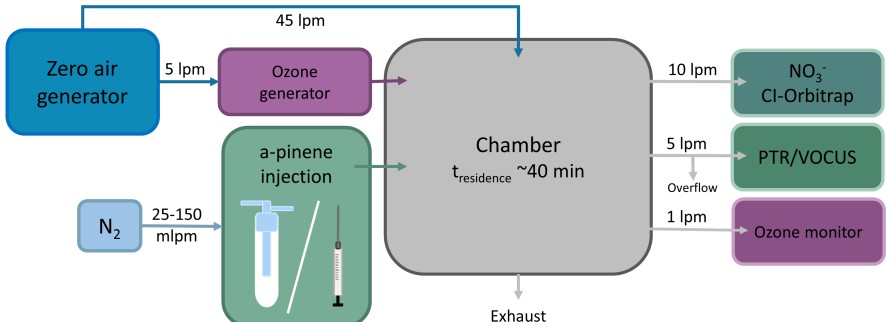

**Figure 3.** The experimental set-up consists of a Continuously Stirred Tank Reactor (CSTR). Additionally, there are ozone and zero air generators, an $\alpha$-pinene injection set-up, an ozone monitor, a nitrate CI-Orbitrap, and a PTR or a Vocus.

## 3.2 Data analysis

We preprocessed the raw data generated by the CI-Orbitrap using a new software called Orbitool (Orbitool version 1.40) developed specifically for the data-analysis of time series data from Orbitrap (Cai et al., 2021). We denoised, mass calibrated, and time-averaged the data into 30-minute steps using Orbitool. We then identified peaks corresponding to HOMs found in the atmosphere (Ehn et al., 2012) and exported the time series of these signals for further analysis. In this work, the term "HOM" includes all compounds with compositions $C_{9-10}H_{14-16}O_{\geq 7}$ and $C_5H_6O_7$ for HOM monomers, and $C_{19-20}H_{28-32}O_{\geq 11}$ for HOM dimers, dimers being $ROOR'$ accretion products formed from $RO_2$ cross reactions (R3 in Sect. 2.2). Noteworthy, other compositions can be and are classified as HOMs (Bianchi et al., 2019), but in this study we aimed to keep the data set more compact for ease of analysis by only inspecting the most prominent and atmospherically relevant signals.

We preprocessed the PTR-ToF and Vocus-PTR raw data using tofTools, a MATLAB based software package (Junninen et al., 2010). Additionally, we conducted sensitivity calibrations to the PTR instruments to obtain calibration factors, which we used to determine the precursor concentrations from the ion signal. The calibration factors were $15$ cps/ppb for the PTR-ToF used for $D_0$, $^7D_2$, and $^{10}D_3$ precursors, and $180$ cps/ppb for the Vocus-PTR used for the $^3D_1$ precursor.

Because the Orbitrap's sensitivity as a function of signal intensity is nonlinear, we determined our instrument's sensitivity behaviour by analysing the $\alpha$-pinene oxidation products and their natural isotopes following the method of Riva et al. (2020). When determining the instrument sensitivity, we excluded all isotopes that contained deuterium, because using selectively deuterated precursors distorts the signal distributions of the natural hydrogen isotopes. Based on our instrument's sensitivity behaviour, we created a correction function to correct the time series data for the nonlinearity. We also used the sensitivity behaviour to determine CI-Orbitrap's detection limit, that is the signal intensity normalised with the total reagent ion signal intensity above which the signal is $\geq 20$ % of the true intensity. The detection limit is $10^{-6}$cps/cps, and it can be converted to a concentration of $10^4$molecules$\cdot$cm$^{-3}$ by multiplying the signal intensity with the calibration coefficient c, however, the conversion introduces large uncertainties. As calibrations for HOMs remain limited due to lack of suitable standards, we assumed the calibration coefficient has the value $c = 10^{10}$ molecules$\cdot$cm$^{-3}$ as an average of typically reported values (Jokinen et al., 2012; Ehn et al., 2014; Riva et al., 2019c). In order to be detected by the nitrate CIMS, the molecule needs to form a very strong cluster with $NO_3^-$ (Hyttinen et al., 2015), and the majority of HOMs are thus expected to be charged at the collision limit, motivating the use of one single c-value for all HOMs. We estimate a large uncertainty of at least -67 %/+200 % arising from the listed assumptions. However, this uncertainty is of less importance in our study, as absolute HOM concentrations are not used for any of the main conclusions of our work.

Additionally, Riva et al. (2020) found that the relative ion transmission of the Orbitrap mass spectrometer is dependent on the mass of the ion. As we did not separately determine the ion transmission of our instrument during these measurements, we corrected our data assuming the same transmission curve as Riva et al. (2020) determined for their instrument. This assumption generates some uncertainty, however, as we are mainly focused on comparing adjacent ions in the mass spectrum, the transmission is unlikely to make a large difference for the purpose of this study. In contrast, in order to assess the impact of deuteration on the total HOM formation, we did also estimate total HOM yields which could have considerable errors if an

erroneous correction is used. Nevertheless, as we will show, using the correction resulted in HOM yields that better matched those of previous studies, while using no correction caused unreasonably high values. Additionally, the HOM yields already have large uncertainties because of the assumptions on the calibration coefficient.

We calculated the total HOM concentration according to

$$[\text{HOM}] = \frac{\{\text{HOM signal}\}}{\{\text{Total reagent ion signal}\}} c, \tag{2}$$

where {HOM signal} is the sum of found HOM signals, {Total reagent ion signal} is the sum of the measured reagent ion monomer ($NO_3-$) and dimer ($HNO_3 * NO_3-$) signals, and $c$ is the calibration coefficient.

We calculated the HOM yield $\gamma$ from the production and loss terms of HOMs when the concentration of HOMs $[HOM]$ did not change significantly over time (Ehn et al., 2014):

$$\frac{d[HOM]}{dt} = k_1 \gamma [\alpha\text{-pinene}][O_3] - k_{loss}[HOM] = 0, \tag{3}$$

from which we can solve for the yield $\gamma$

$$\gamma = \frac{k_{loss}[HOM]}{k_1[\alpha\text{-pinene}][O_3]} \tag{4}$$

where $k_{loss} = \frac{1}{300}$ s $\approx 3.3 \cdot 10^{-3}$ s is the estimated loss rate for HOMs in the chamber dominated by the loss to the chamber walls (Peräkylä et al., 2020).

## 4 Results and discussion

In the following, we first describe the experimental setup and how the oxidation measurements were conducted (Sect. 4.1). We then describe how to interpret the data in form of mass spectra (Sect. 4.2). In Sect. 4.3, we describe how the results were interpreted and what conclusions could be drawn concerning the involvement of different C-atoms. Finally, in Sect. 4.4, we evaluate the effects of the deuteration on the overall HOM yields.

### 4.1 Overview of experiments

We conducted four experiments with three selectively deuterated $\alpha$-pinenes and standard $\alpha$-pinene, i.e. one experiment per precursor, Fig. 4 showing their time series. As can be seen from Fig. 4, optimal steady-state conditions were not achieved in all cases, in particular when the injection was performed using the overflow setup (Fig. A1a), as was the case for $^7D_2$ and $^{10}D_3$. However, we achieved a rate of change in precursor concentration on the order of 10 % $\cdot$ h$^{-1}$ or less in most cases. Thus, the conditions can be considered stable enough from the perspective of the oxidation chemistry, where HOM formation happens on time scales of seconds and HOM loss on the time scale of minutes. For each experiment, we calculated average mass spectra for time ranges where the $\alpha$-pinene reaction rate was comparable between the four precursors. The selected reaction rates $k_1[\alpha\text{-pinene}][O_3]$ were 0.5ppt $\cdot$ s$^{-1}$ and 0.015ppt $\cdot$ s$^{-1}$, which we will refer to as high concentration $C_{high}$ and low concentration $C_{low}$ samples, respectively. The $C_{low}$ sample corresponds to precursor concentration ranges of $1.3 - 1.7$

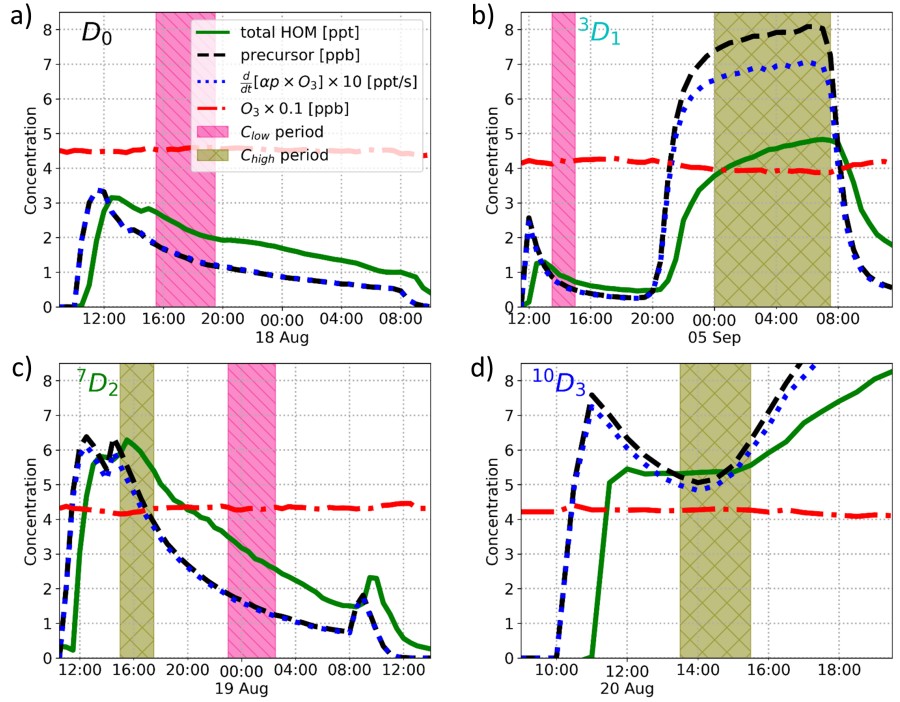

**Figure 4.** Time series of the experiments showing total HOM concentration (green solid line), precursor concentration (black dashed line), $\alpha$-pinene ozonolysis rate multiplied by ten (blue dotted line), and ozone concentration multiplied by 0.1 (red dash-dotted line). Additionally, sample times for low concentration sample ($C_{low}$) and high concentration sample ($C_{high}$) are shown with magenta '\' hatched area and olive cross-hatched area, respectively.

ppb, $0.6 - 1.0$ ppb, and $1.4 - 1.8$ ppb for the $D_0$, $^3D_1$, and $^7D_2$ precursors, respectively. The $C_{high}$ sample corresponds to $7.6 - 8.0$ ppb, $4.6 - 6.0$ ppb, and $5.0 - 5.4$ ppb for the $^3D_1$, $^7D_2$, $^{10}D_3$ precursors.

Due to limitations in the experiments, we could only run one experiment per precursor, which lead to us not achieving a low concentration sample with the $^{10}D_3$ precursor. Similarly, only after the experiment did we conclude that there was no successful $C_{high}$ measurement with the standard $\alpha$-pinene. Thus, we performed the analyses with $C_{high}$ samples when comparing the changes in isotopic distributions due to deuteration, whereas we performed the yield comparisons against the standard $\alpha$-pinene for the $C_{low}$ for all but the $^{10}D_3$ precursor.

We have to consider the purity of the deuterated precursors in the chamber when analysing the data, as it is another source of uncertainty. The standard $\alpha$-pinene has a natural isotope distribution where the D/H ratio is around $0.01$ %, making D-containing isotopes often fall below the detection limit of our instrument. For the selectively deuterated samples, we can inspect their purity using PTR mass spectrometer data. As PTR instruments are known to induce isomerisation and fragmentation of the protonated molecules (Tani et al., 2003; Li et al., 2022), we expect that the observed level of deuteration in the PTR mainly provides a lower limit for the purity. The PTR measurements suggested that over $88$ % of the measured a-pinene in

our chamber contained exactly the number of D-atoms specified, verifying that the samples are very pure and in-line with the observed purity determined using $^1$HNMR spectroscopy (see Sect. 2.1).

The uncertainties arising from the limited number of experiments we could conduct, the estimation of the precursor purity from PTR data, and the estimation of the calibration coefficient in addition to other uncertainty sources discussed later limit the extent to which we can draw conclusions from our data. For this reason, we opt for only broadly discussing the implications of our results, and focusing on detailed mechanistic insights only when they appear unambiguous.

## 4.2   Interpreting the mass spectra

Interpreting the mass spectra requires some thought, and is detailed below for easier reading. Each precursor can form HOM monomers with deuterium atoms up to the number the precursor initially contained. Depending on the formation reactions, the HOMs can have lost deuterium atoms during their formation. As a result, there can be HOM isotopes with the number of deuterium atoms in the molecule anywhere between the number of deuterium atoms in the precursor and zero. Similarly HOM dimers can contain between twice the number of deuterium atoms in the precursor and zero deuterium atoms.

We show an example of the resulting mass spectra for the HOM $C_{10}H_{14}O_{11}$ when using deuterated precursors in Fig. 5. We have gathered the measured spectra from using each deuterated precursor into one figure. In the mass spectra, the HOMs are clustered with the nitrate ion, hence, the nominal mass for $C_{10}H_{14}O_{11}$ when using normal $\alpha$-pinene ($D_0$) is 372 Th (310+62). We omit the reagent ion from all labels for clarity.

    The HOM monomers formed from $^{10}D_3$ precursor can have from zero to three deuterium atoms. As one can see from the
Fig. 5 where $^{10}D_3$ signal is shown with blue bars, we found isotopes containing three, two, and one deuterium atom in the case of $C_{10}H_{14}O_{11}$, and no signal for the compound with zero deuterium atoms. These three found isotopes lie on the mass spectrum 1 Th apart, at m/z 373 Th, 374 Th, and 375 Th, their chemical formulae being $C_{10}H_{13}DO_{11}$, $C_{10}H_{12}D_2O_{11}$, and $C_{10}H_{11}D_3O_{11}$, respectively. Furthermore, for $^{10}D_3$ precursor, the highest signal by a large margin is $C_{10}H_{12}D_2O_{11}$, the HOM that has lost one deuterium atom.

For the $^7D_2$ precursor shown with green bars, we can find $C_{10}H_{12}D_2O_{11}$ at 374 Th, $C_{10}H_{13}DO_{11}$ at 373 Th, and $C_{10}H_{14}O_{11}$ at 372 Th (Fig. 5). The highest signal with a large margin is $C_{10}H_{12}D_2O_{11}$, but in contrast to the $^{10}D_3$ precursor, this HOM has lost zero deuterium atoms during its formation process. Assuming the deuteration does not affect the (preference of) reaction pathways significantly, these findings suggest that in the main pathway forming $C_{10}H_{14}O_{11}$ C-7 carbon is inactive, whereas the C-10 is active. We emphasise that we only consider the carbon to be active when we can observe the D-abstractions take
place at fractions above our uncertainty margins.

## 4.3   HOM isotopic distributions between deuterated precursors

Both radicals and closed-shell products can be found from the $\alpha$-pinene ozonolysis HOM spectrum. Unfortunately, many of the inspected radical signals were covered under large background and oxidation product signals, making it impossible to say whether the radical signal was completely missing or just hidden under the dominant adjacent peak. For example, the
radicals $C_{10}H_{15}O_8$ and $C_{10}H_{14}DO_8$ were typically not distinguishable due to very large signals of the closed-shell species

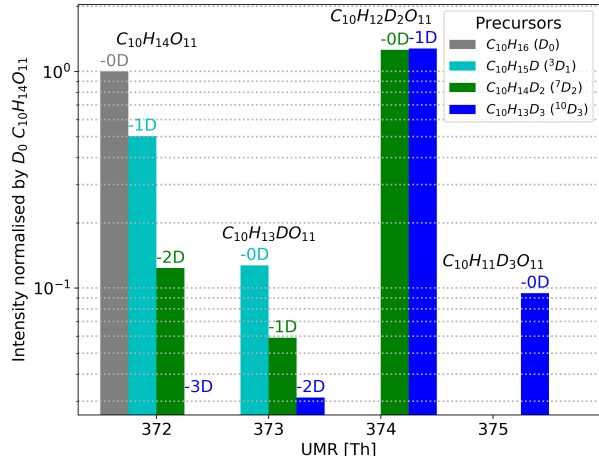

**Figure 5.** Example spectra using $C_{10}H_{14}O_{11}$ for showing resulting HOM signals when using different selectively deuterated precursors. The nitrate cluster of $C_{10}H_{14}O_{11}$ ($C_{10}H_{14}O_{11} \cdot NO_3^-$) is at 372 Th. The colours show which selectively deuterated $\alpha$-pinene is the precursor, and the values above the bars in matching colour tell how many deuterium atoms the HOMs have lost compared to the precursor. For example, the $\alpha$-pinene with 3 deuterium atoms ($^{10}D_3$, shown in blue) can form up to four different $C_{10}H_{14}O_{11}$ isotopes in the spectrum. However, in the $^{10}D_3$ precursor's case there is no signal for $C_{10}H_{14}O_{11}$ without any deuterium atoms (-3D).

$C_{10}H_{13}DO_8$ and $C_{10}H_{16}O_8$, respectively. Hence, despite the high resolution of the Orbitrap, the significant difference in signal intensities prevented us from using the radical data for further analysis in this study, instead focusing our analysis solely on the closed-shell molecules. With lower concentrations and/or shorter residence times, the radical signals would be easier to distinguish as they would account for a larger portion of all the signal (Jokinen et al., 2014; Molteni et al., 2019). We emphasise that the complication arises from the deuteration, as in normal experiments the radicals and closed-shell species would be at different integer masses and thus easily distinguishable.

We focused on compounds that Ehn et al. (2012) found to be the main HOMs in monoterpene-dominated regions like the Hyytiälä forestry field station located in the boreal forest. Due to the sheer number of different signals to look at as there are numerous H/D isotopes for every molecule, we limited the most detailed inspection to eight compounds that correspond to the most prominent signals in $\alpha$-pinene ozonolysis HOM mass spectra: four HOM monomers $C_{10}H_{14}O_7$, $C_{10}H_{14}O_9$, $C_{10}H_{16}O_9$ and $C_{10}H_{14}O_{11}$, and four HOM dimers $C_{19}H_{28}O_{11}$, $C_{20}H_{30}O_{14}$, $C_{20}H_{30}O_{16}$ and $C_{20}H_{30}O_{18}$ (Ehn et al., 2014; Molteni et al., 2019).

By limiting our inspection to these compounds and focusing our main analysis on known ozonolysis products, we limit the effect OH reaction products have on our analysis. Firstly, earlier studies have concluded that HOM formation from OH-initiated reactions is a minor channel in $\alpha$-pinene ozonolysis, only on the order of 10 % of the ozone-initiated oxidation (Ehn et al., 2014; Jokinen et al., 2014, 2015). As such, the impact of OH reactions on the data would be barely detectable especially within our high uncertainty range. Secondly, the limited inspection compounds are formed specifically from ozone reactions (Ehn et al., 2014; Molteni et al., 2019), as the OH-derived HOMs would normally have higher hydrogen atom contents and

odd numbers of oxygen atoms. Thus, our main analysis specifically focuses on known ozonolysis products and the effect of
OH reactions is further negated. Thirdly, the likelihood of multi-generation oxidation, i.e. first-generation non-HOM oxidation
products would react with OH to form HOM, is quite low in our setup. This is because the flush-out time of the chamber is
only 40 minutes and time scale of the wall loss of condensable products is shorter yet (on the order of a few minutes), thus
making OH reactions a relatively small sink for the oxidation products in the chamber (Peräkylä et al., 2020; Bianchi et al.,
2019).

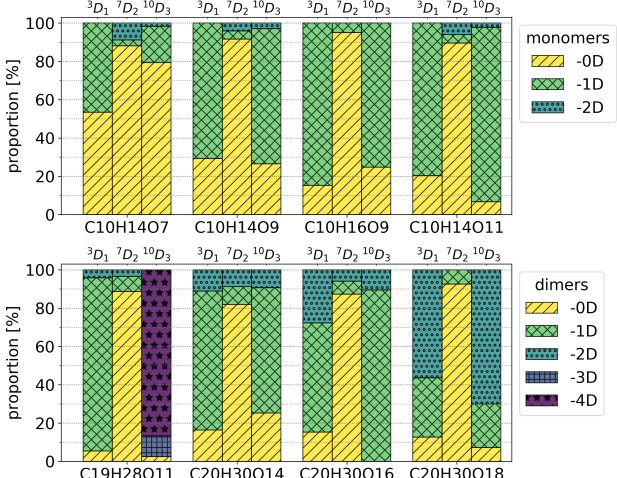

**Figure 6.** Isotopic distributions of chosen HOMs as proportions of signal using high concentration $C_{high}$ data. The colours show how
many deuterium atoms are lost from the molecule during the formation process. For example, in the case of $^3D_1$ precursor and C10H14O7
compound, -0D corresponds to C10H13DO7 and -1D corresponds to C10H14O7. The entries for -3D and -4D are excluded from the upper
legend as there were no corresponding compounds in the inspected monomers.

Isotopic distributions of the four HOM monomers and four HOM dimers for the selectively deuterated precursors are shown
in Fig. 6. Figure A1 shows the same distributions for a much wider range of HOMs, but for these compounds each isotope
was not checked manually, and thus the results are less reliable than the results in Fig. 6 (see appendix for further discussion).
The isotopic distributions are shown for $C_{high}$ data. In general, for the $^3D_1$ and $^{10}D_3$ precursors, the majority of the signal is
generated by isotopes that have lost one or more deuterium atoms, whereas for the $^7D_2$ precursor the signals are predominantly
from isotopes that have not lost any deuterium atoms. In fact, the signal of HOMs that have lost one or more deuterium atoms
for the $^7D_2$ precursor is only at most 10% of the total signal. Due to the large uncertainties in the data, we cannot give an exact
value, but the fraction is clearly so small that the pathways involving abstraction from this carbon are likely not significant. In
other words, the vinylic carbon C-3 and allylic methyl group carbon C-10 are more active in the formation process than the
cyclobutyl ring carbon C-7.
For the more active C-3 and C-10 carbons, we can also note a clear trend for the $C_{10}H_{14}O_x$ monomers and the $C_{20}H_{30}O_x$
dimers, where deuterium loss increases with increasing oxygen content. This makes sense qualitatively, because in order to

reach higher O-atom content more H-shifts need to take place, and almost all potentially abstractable H/D-atoms need to have been abstracted for the most oxygenated HOMs to form. For C-10, the monomers lost at most one D, while the $C_{20}$ dimers lost up to two D, indicating that it is rare to have more than one H/D-abstraction from the same C-atom. The $C_{19}$ dimer is discussed later.

Furthermore, the data for $^3D_1$ and $^{10}D_3$ precursors implicitly show that there are multiple structural isomers contributing to the different elemental compositions. If there was only one isomer, the number of lost D-atoms would be identical for all molecules with a given composition resulting in signal of only one isotope. However, as we can see from Fig. 6 (and Fig. A3), there are signals for multiple isotopes.

We can now compare our results to those of Iyer et al. (2021), where they suggest a mechanism leading to the $C_{10}H_{15}O_8$ $RO_2$ radical shown in Fig. 2b. More specifically, we can look at the closed-shell product $C_{10}H_{14}O_7$ formed from the $O_8-RO_2$ through OH-loss and compare its expected behaviour based on the proposed mechanism to the behaviour observed when using selectively deuterated samples. According to the mechanism proposed by Iyer et al. (2021) (Fig. 2b), we see that out of the deuterated positions used in this work (C-3, C-7, and C-10), only C-3 has undergone an H-shift (cyan H). Hence, we would expect that the closed-shell $C_{10}H_{14}O_7$ HOM would always have lost D through exchange with water (Fig. 2c) when using $^3D_1$, while the HOM would never have lost a D when using $^{10}D_3$ and $^7D_2$ precursors. However, according to our observations, for $^3D_1$ signal of $C_{10}H_{14}O_7$, roughly half still has the D attached (Fig. 6), meaning that the Iyer et al. (2021) pathway can at most explain half of the observed signals in our experiment. For the $^{10}D_3$ $C_{10}H_{14}O_7$, about 20 % has lost a D-atom, while the other 80 % behaved according to expectations from Iyer et al. (2021). However, the Iyer et al. (2021) mechanism is not necessarily the only mechanism through which all of the 80 % of the signal is formed, as there can be other mechanisms that could explain parts of the signal.

Lastly, the $C_{19}H_{28}O_{11}$ dimer is often the most abundant dimer measured by the nitrate CIMS from $\alpha$-pinene ozonolysis (Ehn et al., 2014; Rissanen et al., 2015) and also a dominant signal in night-time atmospheric measurements (Yan et al., 2016), but no suggestions have been made concerning which C-atom is lost. In Fig. 6, the signal of this dimer for $^{10}D_3$ precursor has always lost at least three D, providing clear evidence for C-10 being the one that is lost. This is also true more generally for $C_9$ and $C_{19}$ HOMs according to Fig. A3, where we can see a clear difference in the isotopic distributions for the $^{10}D_3$ precursor, namely the majority of the $C_9$ and $C_{19}$ signals lost all three D-atoms.

To summarise the findings from our approach of studying selectively deuterated $\alpha$-pinene, the observed roles of different C-atoms in the autoxidation process are highlighted in Fig. 7. The carbons C-3 and C-10 (green stars) are active and often lose deuterium atoms during the HOM formation process. Additionally, the fraction of signal that had lost deuterium atoms increased with the increasing oxygen number of the HOM. In contrast, the carbon C-7 (cyan diamond) rarely lost deuterium atoms meaning it was mostly inactive in the HOM formation process. As for the carbons C-2 and C-6 (grey hexagons), they can neither be active as defined in this work, nor studied with selective deuteration because they are not attached to any hydrogen atoms. The carbons C-1, C-4, C-5, C-8, and C-9 were not inspected in this study, but selectively deuterated studies of these would be extremely interesting, and provide crucial information for understanding the detailed autoxidation and HOM formation mechanisms of this system.

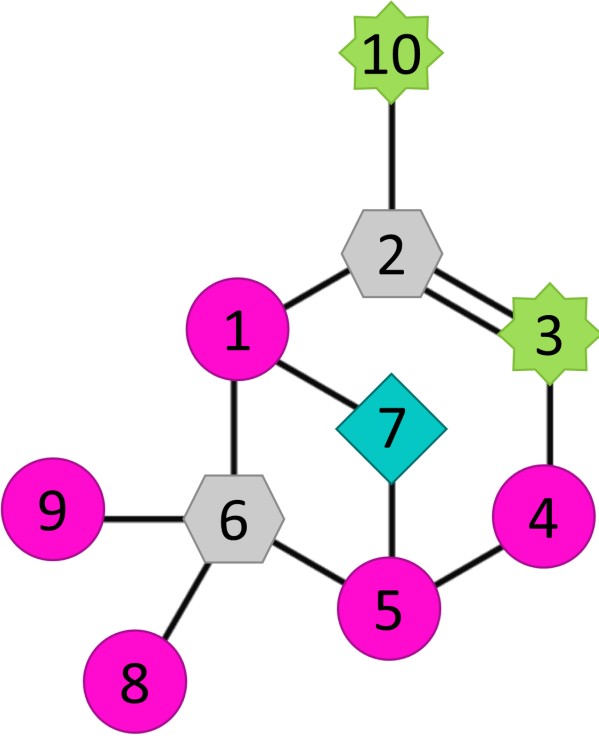

**Figure 7.** Molecular structure of $\alpha$-pinene with numbered carbons showing carbons that are active (carbons 3 and 10; green 8-point star) and inactive (carbon 7; light blue diamond) in the formation process, carbons that were not inspected in this study (carbons 1, 4, 5, 8, 9; magenta circle), and carbons that cannot be studied using selective deuteration (carbons 2 and 6; grey hexagon).

## 4.4 HOM yields

Until now, we have only looked at the relative isotope distributions of specific elemental compositions from separate precursors. However, as described in Sect. 2.3, the deuteration may also slow down the autoxidation, which could result in an overall
375    decrease of HOM yields. In this section, we inspect how selective deuterations affected the yields of different HOMs in comparison with HOM yields from standard $\alpha$-pinene ozonolysis.

Following the definition of HOM in Sect. 3.2, the total HOM yields (Fig. 8a) are calculated from the sum of the signals of all found HOMs for the $C_{low}$ samples, except for the $^{10}D_3$ precursor for which the $C_{high}$ sample was used. Not all HOM peaks were necessarily found, especially in the lower masses and with lower signal intensities due to the high background and
380    instrumental limitations. Nevertheless, after correcting the data for the CI-Orbitrap's relative ion transmission and nonlinearity of the sensitivity as a function of signal intensity, the HOM yield of the $D_0$ precursor is roughly 5 %, which is similar to values reported earlier for $\alpha$-pinene ozonolysis experiments which ranged from 1.2 (+1.2/-0.72) to $7.0 \pm 3.5$ % (Bianchi et al., 2019). Regardless, we again emphasise that there are large uncertainties involved in determining this absolute value, and we

are mainly interested in the possible change of the HOM yields with different precursors. This comparison should be much more robust than the determination of absolute HOM yields.

We report the HOM yields grouped by their chemical compositions in Fig. 8a due to their differing behaviour. $C_5H_6O_7$ is separated due to the signal being much higher than the rest of the HOM signals. The yields of HOM monomers with 7 O-atoms and HOM monomers with 8 or more O-atoms were separated due to the potentially differing volatilities between the two groups of molecules, the $O_7$ monomers having a higher volatility (Peräkylä et al., 2020). Lastly, we give one yield value to the rest of the HOMs, namely the HOM dimers.

Surprisingly, the selective deuterations used in this study did not decrease the HOM yields significantly. We can see in Fig. 8a the obtained total HOM yields for precursors $^3D_1$, $^7D_2$, and $^{10}D_3$ were 3.7%, 6.3%, and 3.5%, respectively. Considering the instrumental uncertainties, and the differences in precursor concentrations between the experiments, we conclude that the $\pm 20 - 40$ % variability is within the uncertainty limits of our experiment. However, this change can be compared to the change of two orders of magnitude observed by Rissanen et al. (2014) for fully deuterated vs non-deuterated cyclohexene, suggesting that deuteration had a minimal role, if any, for the HOM yields of our precursors.

Furthermore, the detailed spectra from the deuterated precursors were similar to that of standard $\alpha$-pinene. The compound-wise comparison of HOMs between $D_0$ and deuterated precursors are shown in Fig. 8b-d. We calculated the compound-wise yields by using the sum of all found isotope signals corresponding to a specific HOM composition. As an example, to calculate the compound-wise yield for $C_{10}H_{14}O_{11}$ for the $^{10}D_3$ precursor, we take all the found isotope signals corresponding to $C_{10}H_{14}O_{11}$ (blue bars in Fig. 5), sum them, and use that sum to calculate the yield for the specific compound. What is more, we limited the inspection in Fig. 8b-d to the compounds that were found in both the deuterated precursor's spectrum and $D_0$ precursor's spectrum. As can be seen from the figures, the data points fall very close to the 1:1 line in all cases, meaning that the total concentrations of HOMs in the deuterated precursors' spectra do not differ significantly from the concentrations of HOMs formed from the $D_0$ precursor. Additionally, there is no clear trend that more oxygenated HOMs would be more impacted by the deuterations, although we showed that they typically had lost more deuterium atoms. This may be an indication that the D-shifts were still fast enough to outcompete other reaction pathways despite the deuteration. On the other hand, it is possible that the autoxidation could proceed through the next most competitive pathway not shut down by the deuteration and end up losing deuterium atoms later in the process, especially in the case of the more oxidised products.

The compound-wise HOM yields for $^{10}D_3$ are in general slightly lower than the yields for $D_0$ precursor (Fig. 8d), which is mainly caused by us comparing the high concentration $C_{high}$ $^{10}D_3$ to the low concentration $C_{low}$ $D_0$ sample. We saw a similar decrease in HOM yields for the other two selectively deuterated precursors when using $C_{high}$ data which may be explained by higher precursor loadings that decrease the $RO_2$ lifetimes, thus hampering the HOM formation. In other words, comparing the $C_{high}$ $^{10}D_3$ to the $C_{low}$ $D_0$ sample would only exaggerate the HOM yield decrease we expect to see from the deuteration. As the difference is still small enough, we can conclude that the decrease caused by selective deuteration is not significant for the $^{10}D_3$ HOMs either, matching what was found for the other two selectively deuterated precursors.

Noteworthy, the $O_7$ yield in Fig. 8a is smaller for the $^3D_1$ sample. A difference can also be seen in the mass spectra (Fig. A2), namely the masses 308 ($C_{10}H_{14}O_7$) and 310 ($C_{10}H_{16}O_7$) Th have markedly lower signals in the spectrum for $^3D_1$ compared

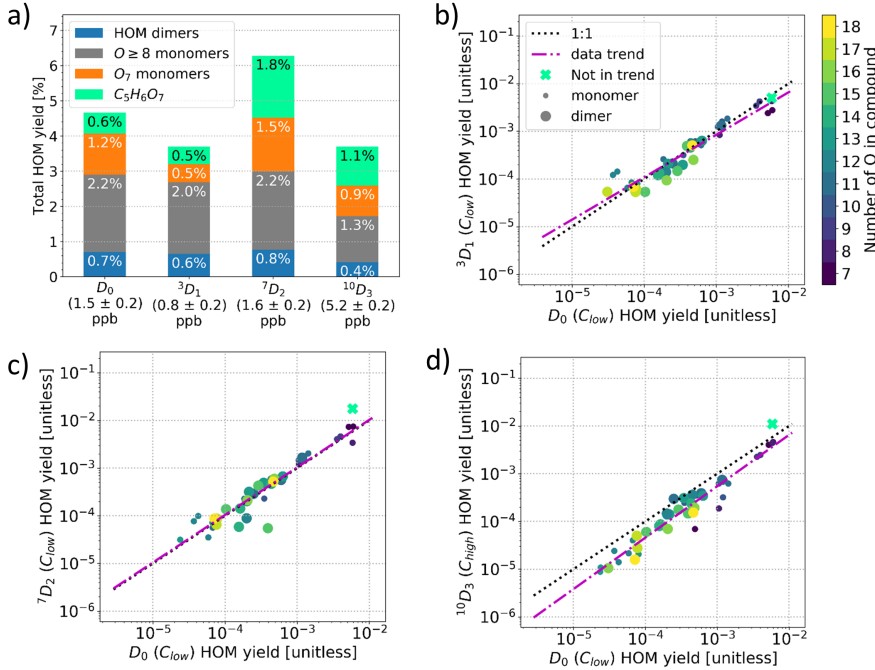

**Figure 8.** a) HOM yields for $\alpha$-pinene samples, where yields for $C_5H_6O_7$ (mint green), $O_7$ monomers (orange), $O_{8+}$ monomers (grey), and dimers (blue) are plotted separate. The corresponding precursor names and concentrations are shown on the x-axis. b-d) Comparison of HOM yields compound per compound between $D_0$ sample and $^3D_1$ sample (b), $^7D_2$ sample (c), and $^{10}D_3$ sample (d), where the mint green cross corresponds to $C_5H_6O_7$ and is excluded from the data trend, and the colour shows the number of oxygen atoms in the compound. $C_{low}$ and $C_{high}$ refer to the $\alpha$-pinene concentration of the sample. NOTE each point in b-d) corresponds to the yield of the sum of all found isotopes per main compound.

to the $D_0$ spectrum. Figure A3 shows that for $C_{10}H_{14,16}O_7$ there is a clear difference in fraction of D lost between $^3D_1$ and
420   $^{10}D_3$, but at $C_{10}H_{14,16}O_9$ the difference is gone. Likewise, the yields of HOM with 8+ O-atoms (grey bars in Fig. 8a) are very close between the different labelled compounds. From a $RO_2$ radical perspective, it is clear that to form $C_{10}H_{15}O_{10}$, one necessarily has gone through $C_{10}H_{15}O_8$, and as such it is surprising that the products of the less oxidised radicals can decrease while the more oxidised products do not. It is possible that different structures of $C_{10}H_{15}O_8$ can form, and the ones that can undergo further H-shifts are not hampered by deuteration, while the isomers that cannot undergo H-shifts and thus will
425   terminate to e.g. $C_{10}H_{14}O_7$ are hampered. Based on the data we have, the latter option is supported, however, without data on the radicals this finding remains highly speculative.

# 5    Conclusions

In this study, we tested the method of selective deuteration for assessing autoxidation mechanisms that are otherwise challenging to elucidate. We monitored the highly oxygenated oxidation products of three selectively deuterated $\alpha$-pinenes to calculate the fractions of HOM that had lost D-atoms. We found that in the cases where carbons C-3 and C-10 had been deuterated, D-atoms were frequently lost, indicating that HOM formation happens through H-abstractions from these C-atoms. Contrastingly, no significant D-atom loss was observed in the case where C-7 had been deuterated, suggesting that this C-atom was rarely involved in any isomerisation reactions during autoxidation and HOM formation. Our findings not only support the existence of the channel Iyer et al. (2021) postulated, but also indicate that there are additional, so far unknown oxidation channels for HOM formation from $\alpha$-pinene ozonolysis. Furthermore, we found that the used deuterations did not significantly affect the HOM yields contrasting our expectations based on kinetic isotope effects.

Our experiments demonstrate that selective deuteration can be a very powerful tool for identifying autoxidation mechanisms. What is more, we can identify many improvements to these initial investigations. Firstly, the interpretation of the results would be simpler if we were able to observe the radicals themselves. Even better still would be to monitor less oxygenated radicals, which has so far been shown to be possible only using few different instruments (e.g. Berndt et al., 2018). The radical signals are more straightforward to interpret because these compounds are at an earlier stage of the oxidation processes, removing different types of branching possibilities that can take place when the inspected closed-shell products are formed. Unfortunately, due to the large background and overlapping isotope signals that possibly obfuscated some of the signals at the smaller masses, we were not able to inspect the radical signals with the experimental design we used in this study.

Secondly, for a more complete picture of the autoxidation pathways, a more comprehensive set of selectively deuterated compounds would be beneficial. Lastly, it might be possible to investigate the most prominent signals with tandem mass spectrometry (MS$^2$) and gain further insight into their structures (Tomaz et al., 2021).

We have shown that using selectively deuterated precursors for studying the ozonolysis and oxidation of $\alpha$-pinene is a viable method for mechanism development to gain insight into the autoxidation process still veiled in limited understanding of exact reaction pathways. We expect that utilising this approach and possibly other related isotope-based approaches will be the most promising route forward for understanding autoxidation propagation on a mechanistic level, both for $\alpha$-pinene and other complex atmospheric VOCs of importance. As such, we are planning a continuation study with a more mechanistic orientation.

*Data availability.*    Data is available upon request. Data presented in the manuscript is archived in Zenodo (DOI: 10.5281/zenodo.7756182; Meder et al., 2023).

 **Appendix A**

## A1 $\alpha$-pinene injection set-ups

The schematics of the $\alpha$-pinene injection set-ups used in the experiments are shown in Fig. A1. The overflow injection set-up shown in Fig. A1a) consists of a gas bubbler and a small glass vial containing a very small amount (few tens µl) of the precursor. The glass vial is covered with parafilm with a small hole in the film to control the rate at which the precursor is flushed out. A small flow of $N_2$ (25-150 mlpm) is flushed over the glass vial through the gas bubbler to carry the evaporated precursor into the chamber. Due to the very small amounts of precursor samples we had, we wanted to use all the precursor sample from the vial. Thus, we added also the cap of the glass vial inside the bubbler in case any of the precursor was on it. This set-up was used when there was not enough precursor to take a sample in a syringe, and the resulting precursor concentration in the chamber was less stable than with the syringe pump injection set-up.

The syringe pump injection set-up shown in Fig. A1b) was used when there was enough precursor to take a sample with a 5 µl syringe. We used a syringe pump to inject the precursor into a stream of $0.5$ lpm $N_2$ flow carrying the precursor into the chamber. The precursor is injected into the stream using a small T-piece, having one of its openings sealed with a septa plug through which the syringe is put, and the two others connected to the $N_2$ flow. This set-up was much more controllable than the overflow set-up, however, there was a communication problem between the pump and the controlling software which caused the pump to inject at an order of magnitude faster rate within a certain range of set injection rates. In some cases, this resulted in far too high precursor concentrations in the chamber, increasing the large background and prohibiting us from using that data for analysis.

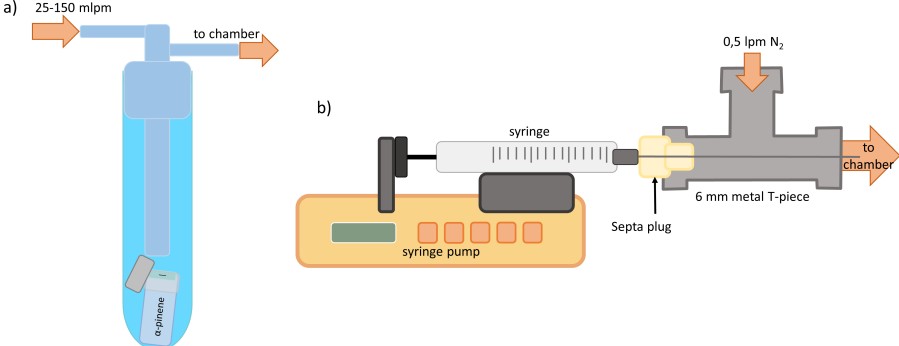

**Figure A1.** $\alpha$-pinene injection set-ups used in the experiments. a) The overflow injection set-up was a gas bubbler into which the small glass vial holding the precursor sample (and its cap) were inserted. b) The syringe pump injection set-up consists of a syringe pump, a syringe, a metal T-piece and a septa plug.

## A2    HOM mass spectra

Figure A2 shows the mass spectra for all the precursors used, highlighting the closely inspected compounds and their isotopes
used in the Fig. 6, i.e. four HOM monomers $C_{10}H_{14}O_7$, $C_{10}H_{14}O_9$, $C_{10}H_{16}O_9$ and $C_{10}H_{14}O_{11}$, and four HOM dimers
$C_{19}H_{28}O_{11}$, $C_{20}H_{30}O_{14}$, $C_{20}H_{30}O_{16}$ and $C_{20}H_{30}O_{18}$.

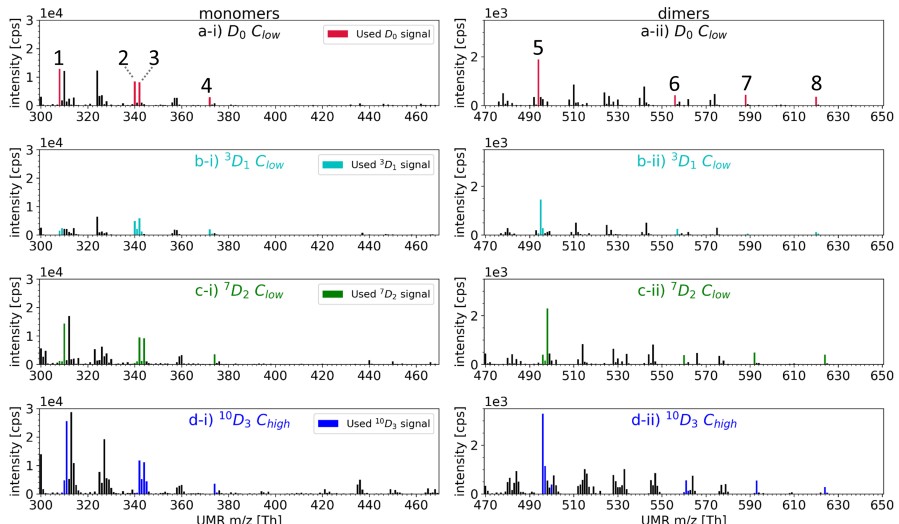

**Figure A2.** Mass spectra of HOMs from selectively deuterated $\alpha$-pinene precursors in unit mass resolution. Masses corresponding to HOM
signals used in closer inspection are marked with a colour, whereas other peaks in the mass spectra are shown in black. a-i and a-ii) The closer
inspection compounds are 1. $C_{10}H_{14}O_7$, 2. $C_{10}H_{14}O_9$, 3. $C_{10}H_{16}O_9$, 4. $C_{10}H_{14}O_{11}$, 5. $C_{19}H_{28}O_{11}$, 6. $C_{20}H_{30}O_{14}$, 7. $C_{20}H_{30}O_{16}$, and
8. $C_{20}H_{30}O_{18}$. The y-scales are different between the monomer and dimer plots.

## A3    HOM isotopic distributions for all found HOMs

Figure A3 shows the isotopic distributions for all found HOMs, including the compounds shown already in the Fig. 6. Note-
worthy, unlike the signals for the compounds in Fig. 6, most of these signals have not been checked for overlapping signals,
hence we cannot say for sure whether there was no signal at all for some isotopes or that some of the signals were obfuscated
by larger nearby signals. Additionally, since many of these signals are clearly smaller than those in Fig. 6, it is possible that
only one of the isotopes' signals is above the detection limit of our instrument, which would lead to a 100 % contribution from
that isotope, although others would have been almost equally high. Therefore, caution should be taken when interpreting the
presented distributions.

We can see that $C_9$ and $C_{19}$ compounds have usually lost at least three deuterium atoms when using the $^{10}D_3$ precursor. In
other words, it is likely that the carbon lost in the formation of the $C_9$ and $C_{19}$ compounds is C-10, fragmenting with all the

deuterium atoms connected to it. The behaviour differs significantly from that of $C_{10}$ and $C_{20}$ compounds, who have at most lost two deuterium atoms.

Additionally, when using the $^3D_1$ precursor, the $C_9$ compounds' signals consist mostly of the isotopes that have lost 1 deuterium atom, and $C_{19}$ compounds mostly lost 1 deuterium and sometimes 2 deuterium atoms, meaning that the C-3 carbon is active in the formation process of these compounds. In contrast, these compounds' signals consist mostly of the isotopes that have lost zero deuterium atoms when using the $^7D_2$ precursor, meaning that the C-7 is rarely active.

Figure A3 also contains the isotopic distributions for $C_5H_6O_7$, for which the carbons C-3 and C-10 have in practice always lost all of the deuterium atoms and C-7 has never lost any. This likely means that of the five carbons lost from $\alpha$-pinene when forming this compound, two of them are the C-3 and C-10 carbons.

Furthermore, we can see the general trends depicted in Fig. 6 also in Fig. A3, where C-3 and C-10 are active and C-7 is inactive in the formation process.

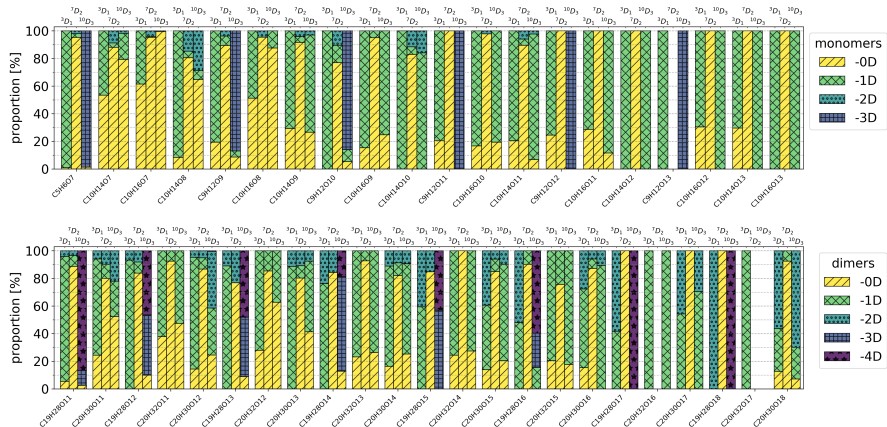

**Figure A3.** Isotopic distributions of all inspected peaks. The colour shows how many deuterium atoms have been lost from the precursor during the formation of the HOM isotope that corresponds to the main HOM on the x-axis. For example, in the case of $^3D_1$ precursor and C10H14O7 compound, -0D corresponds to C10H13DO7 and -1D corresponds to C10H14O7. The entry for -4D is excluded from the upper legend as there were no corresponding compounds in the inspected monomers.

## A4    HOM yield behaviour as a function of average number of D lost

Figure A4 show the comparison of HOM yields compound per compound between D0 precursor and the deuterated precursors. It is otherwise the same as Figs. 8b-d, except the colour shows the average number of deuterium atoms lost per compound. This modification could highlight the effect of the kinetic isotope effects better, however, despite the modification we do not see a clear effect from the kinetic isotope effects. Nonetheless, Fig. A4 highlights the differing behaviours of the precursors, i.e. how $^7D_2$ HOMs mostly did not lose deuterium atoms, whereas $^3D_1$ and $^{10}D_3$ have (the colour scales have a larger range for the latter two precursors). Additionally, we can see that for $^3D_1$ precursor, most of the HOMs have lost on average some

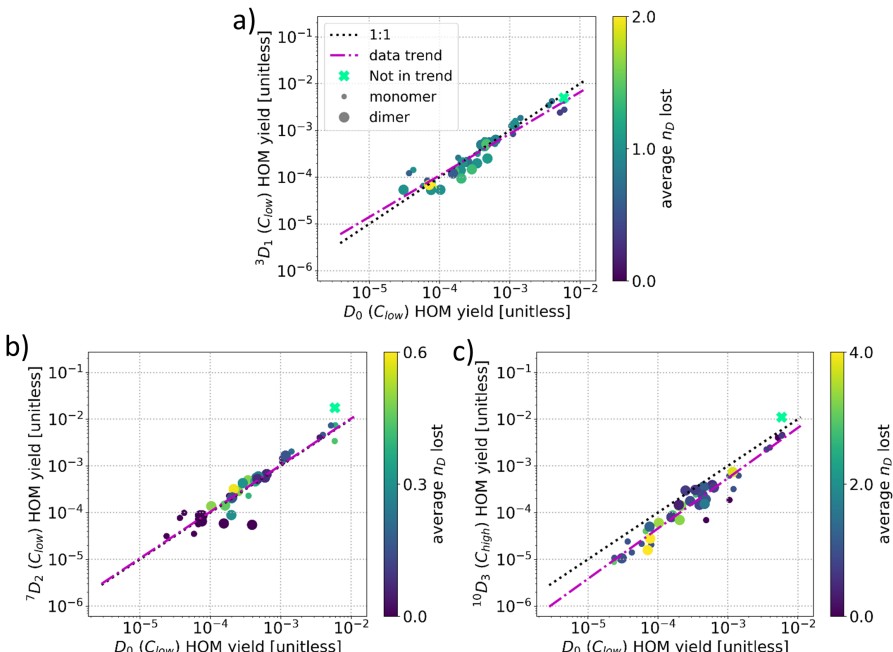

**Figure A4.** Comparison of HOM yields compound per compound with the number of deuterium atoms lost on average shown with the colour between $D_0$ sample and $^3D_1$ sample (a), $^7D_2$ sample (b), and $^{10}D_3$ sample (c), where the mint green cross corresponds to $C_5H_6O_7$ and is excluded from the data trend. $C_{low}$ and $C_{high}$ refer to the $\alpha$-pinene concentration of the sample. NOTE each point corresponds to the yield of the sum of all found isotopes per main compound.

deuterium atoms (most points are in the green colour region), whereas for $^{10}D_3$ precursor HOMs, there are clearly some HOM that have lost more deuterium atoms than others (some points are yellow, where others are in the blue to green colour region).

*Author contributions.* ME and RJT conceived of the overall project idea. JL and JV synthesised the deuterated pinene precursors. MM and ME planned the experiments, and MM conducted the experiments, carried out the data analysis, and wrote the paper under the guidance of ME. All authors discussed the results and commented on the paper.

*Competing interests.* The authors declare they have no conflicts of interest.

*Acknowledgements.* We thank Pekka Rantala, Hannu Koskenvaara, and Petri Keronen from the technical staff at INAR/Physics at the University of Helsinki for the indispensable help and assistance during the experiments. Additionally, we thank Frans Graeffe for calibrating and running the Vocus PTR-ToF, and Yihao Li for his efforts in developing the software tool Orbitool for CI-Orbitrap data analysis.

Financial support. This project has received funding from the European Research Council under the European Union's Horizon 2020 research and innovation programme under Grant No. 101002728, the Academy of Finland (Grant Nos. 317380, 320094, 331207), the National Science Foundation, USA under Grant No. CHE-1607640 (to RJT), and the Magnus Ehrnrooth foundation.

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
