# Peer review of "Selective deuteration as a tool for resolving autoxidation mechanisms in $\alpha$ -pinene ozonolysis"

_EGUsphere, 2022_

## Referee Comment (RC2)

Review of "Selective deuteration as a tool for resolving autoxidation mechanisms in α-pinene ozonolysis" by Meder et al., (egusphere-2022-1131)

In this manuscript, Meder et al. investigated ozonolysis of normal and partially deuterated α-pinene, focusing on the distribution of highly oxygenated molecules (HOMs) with different numbers of D atoms and HOM yields when different partially deuterated were used. This technique of utilizing partially deuterated precursors indeed will help to better understand reaction pathways, but the authors should have gone much further beyond their current discussion. Especially, they are advised to connect their observation to potential mechanisms, and at least try to suggest the formation routes of some of the most abundant products. I would recommend major revision of this manuscript.

Main concerns,

1. The authors seem to overlook OH radical formation in the ozonolysis of α-pinene. In fact, the OH yield is pretty high, which leads to a dispensable fraction of OH+α-pinene in this system. I mean, then the observed distribution of (partially deuterated) HOMs does not necessarily reflect the ozone chemistry. Also, OH might react with the HOM products, further altering their distribution. How to evaluate the impact of secondary reactions on HOM distribution? By the way, why wouldn't the authors give the extent of consumption of precursors in each case?

2. It was stated that "autoxidation was perturbed in predictable ways by the deuteration" (Line 40, Page 2). On the other hand, there is only one paragraph that discussed confirmation of a detailed reaction pathway by Iyer et al. 2021. The rest just ends up with "C-3 and C-10 are active and C-7 is mostly inactive". If they can exclude potential interferences from OH+α-pinene and OH+HOMs as I raised above, the authors might be able to figure out routes beyond the Iyer et al. 2021 study.

3. Figure 6 is confusing to me. Please at least give one example of the definition in the caption. For example, in the case of $C10H14O7$, when $^3D_1$ was the precursor, -0D corresponds to $C10H13DO7$ and -1D corresponds to $C10H14O7$? When $^7D_2$ was the precursor, -0D corresponds to $C10H12D2O7$, -1D corresponds to $C10H13DO7$, and -2D corresponds to $C10H14O7$?
   Why aren't there a -3D in the upper panel of Figure 6? Do the authors suggest loss of 3D in $^3D_1$ was not observed?
   The color scale in Figure 6 might be modified for a better differentiation.

Minor comments

4. (Paragraph 2, Page 2), selective deuteration has been used to study autooxidation mechanisms in the 1,3,5-trimethylbenzene + OH system (Wang et al., ACP 2020, 20, 9563-9579), which is highly relevant to this study.

5. The author states that "the purity of each compound was >95%" in line 66, and that "PTR measurements suggests that over 88% of the measured pinene in our chamber contained exactly the number of D-atoms specified" in Line 218. The two numbers are close, but could still lead to a problem for the purpose of a mechanism study. Please revise or justify.

6. "Inactive" is not a perfect word to describe whether or not a D atom goes through autooxidation. Since this manuscript judge by the distribution of products, but even for the "inactive" precursor with selective deuteration at the C-7, products with loss of D atoms were observed, which suggests that something happened to the C-D bond.

7. (Line 302), one can also argue that $C_{19}H_{28}O_{11}$ can be formed from a $^3D_1$ precursor, since the signal of $C_{19}H_{28}O_{11}$ in the case of a $^3D_1$ precursor could be attributed to mostly loss of a D atom. Does this mean that $C_{19}H_{28}O_{11}$ is formed via one monomer that reacts at C-3 and the other at C-10?

---

## Referee Comment (RC3)

**Selective deuteration as a tool for resolving autoxidation mechanisms in $\alpha$-pinene ozonolysis**

**Major Comment**

The first thing I would highlight is the effort gone into presenting this work in a digestible way. The colour coding of the deuteration, and then presenting the reaction routes with this colour coding makes working out how the products are potentially formed straightforward.

The big picture results of this study is that vinylic C-3 and allylic methyl C-10 carbons are active but the cyclobutyl ring carbon, C-7 is not. Via the C-3 isomer, ~ 50% of its loss is consistent with the literature Iyer *et al*. mechanism. In general, only one D loss occurs in the oxidation process, but there is a mechanism where the CH3 is lost, and strong evidence for this is seen via the $^{10}D_3$ isomer data. Regarding the total HOM yield, there is little difference between the deuterated isomer (and normal alpha-pinene) and is around the 5% yield; but the $^7D_2$, which is inactive, does have the highest yield. However, the lack of significant kinetic isotope for the active $^3D_1$ and $^{10}D_3$ is surprising. There probably should be a little bit more about potential mechanisms that produce little or no kinetic isotope effect. Is there a predicted isotope effect via the Iyer *et al.* mechanism?

However, things are probably simplified too much, and while I like this approach there should be acknowledgement of the extra stuff going on. The most obvious is that at the early stage of ozonolysis a significant amount of OH is made, and this is going to react with the pinene to make different peroxy radicals that are going to undergo auto-oxidation leading to HOM formation. You note that for $^3D_1$ via the Iyer mechanism accounts for about 50% of the signal. Could it be that OH chemistry is accounting for this missing signal?

While I might be showing my ignorance of the experimental details, as I understand it a slow flow of O3/alpha-pinene is introduced into the chamber and it typically takes about 40 minute to flow out. So the experiment is essential a snap shot in time at 40 minutes. However, the system never seems to reach this 40 minute snap shot in time, and the products are evolving over a much longer timescale. Does this indicate that the surface of the chamber is playing a role in these experiments? If this is the case, can you be sure that the products are the result of only gas-phase chemistry. I think some acknowledgement of the role of the chamber surface is required. Or can it be dismissed?

Overall, this work has demonstrated that via deuteration of the target alkene some of the HOMs that form via ozonolysis can assigned to specific isomerization steps. This is useful information. There are a number of things that can be done to put this work into better perspective, but overall I have no problem recommending this paper for publication.

**Specific comments**

**Line 80**
*The O8−RO2 product remains the only experimentally and computationally supported HOM-forming pathway in this system*

This mechanism involves the $^3D_1$ bond breaking. Then this route should show a kinetic isotope effect? Is there a number from theory?

**Line 130**
*if the H-shift is 1000 times faster than any competing reaction, substituting for a D-shift will barely impact the branching of the reaction,*

While true about the branching ratio, but it will take longer to form if deuterated, hence yields should be lower, certainly at early times.

**Line 145**
*We injected _-pinene into the chamber using one of two methods: an overflow set-up for small amounts of precursor, and a syringe pump set-up when there was enough precursor for using a 5 µl syringe to take a sample. In both cases, the evaporated_-pinene was injected into the chamber with a small N2 flow.*

So what was the typical [pinene] in the chamber at the start? I can see the answer to this in Figure 5. Perhaps it should be stated in the experimental section.

**Line 177**
*We excluded all isotopes that contained deuterium, because using selectively deuterated precursors distorts these signals.*

Is this exclusion only for determining the instruments sensitivity?

**Line 199**
*We calculated the HOM yield from the production and loss terms of HOM when the concentration of HOM [HOM] did not change significantly over time (Ehn et al., 2014):*

But your experiments have a constant residence time in the chamber, so what you means by "time"

**Line 223**
*Similarly HOM dimers can contain between twice the number of deuterium atoms in the precursor and zero deuterium atoms.*

There needs to be more on HOM dimers. Perhaps introduce them in section 2.2. What is a typical HOM dimer yield in this study? Are they at steady-state?

**Line 242**

*As can be seen from Fig. 5, optimal steady-state conditions were not achieved in all cases, in particular when the injection was performed using the overflow setup (Fig. A1a), as was the case for 7D2 and 10D3.*

Again, as the time in the chamber of constant, i.e. time before measurement, where is this change with time coming from? I presume it is linked to things going to the walls before the walls are at steady-state, which is not the same as the steady-state of equation (2). If you run a model of the system, what time does this predict for steady-state, where I presume the model will have loss to the wall as a constant.

**Line 263**

*as in normal experiments the radicals and closed-shell species would be at different integer masses and thus easily distinguishable.*

But you have done "normal experiments" when doing $D_0$. So can you show some radical data? Are the radicals at steady-state?

**Line 299**

*"For the 10D3 C10H14O7, about 20 % has lost a D-atom, while the other 80 % behaved according to expectations from Iyer et al. (2021)."*

80% if no other mechanisms is considered!

**Line 338**

*However, this change can be compared to the change of two orders of magnitude observed by Rissanen et al. (2014) for fully deuterated vs non-deuterated cyclohexene, suggesting that deuteration had a minimal role, if any, for the HOM yields of our precursors.*

It is clear that C—D bonds have been broken in this study, but to observe slight change in the yield is surprising. I think a little more speculation on this observation is required.

**Line 350**

*This may be an indication that the D-shifts were still fast enough to outcompete other reaction pathways despite the deuteration. On the other hand, it is possible that the autoxidation could proceed through the next most competitive pathway not shut down by the deuteration and end up losing deuterium atoms later in the process, especially in the case of the more oxidised products.*

While deuteration might not change reaction paths, it should slow the rate to products? So the second sentence explanation is more likely. Would you like to speculate how the D is happening later in the process and not show a significant kinetic isotope effect?

---

## Author Response (AR1)

**Final Response for "Selective deuteration as a tool for resolving autoxidation mechanisms in $\alpha$-pinene ozonolysis"**

Melissa Meder[1], Otso Peräkylä[1], Johnathan G. Varelas[2], Jingyi Luo[2], Runlong Cai[1], Yanjun Zhang[1], Theo Kurtén[1,3], Matthieu Riva[4], Matti P. Rissanen[5,3], Franz M. Geiger[2], Regan J. Thomson[2], and Mikael Ehn[1]

[1]Institute for Atmospheric and Earth System Research (INAR/physics), University of Helsinki, Helsinki, Finland
[2]Department of Chemistry, Northwestern University, Illinois, USA
[3]Department of Chemistry, University of Helsinki, Helsinki, Finland
[4]Univ Lyon, Université Claude Bernard Lyon 1, CNRS, IRCELYON, Villeurbanne, France
[5]Aerosol Physics laboratory, Tampere University, Tampere, Finland

**Correspondence:** Melissa Meder (Melissa.Meder@helsinki.fi) and Mikael Ehn (Mikael.Ehn@helsinki.fi)

We thank all four referees for their input on our manuscript. We address their comments and concerns below. We start by addressing the three main concerns shared by many of the referees before continuing to the point-by-point responses. Referee comments are quoted in italics, while our responses are given in normal font and are numbered as R1, R2, R3, etc. for easier referencing. Amendments in the manuscript text are given in red.

R1) The referees had concerns that OH+$\alpha$-pinene reactions could be important and impact some of our findings and conclusions. We did not discuss OH+$\alpha$-pinene reactions in our manuscript because many earlier studies (Ehn et al., 2014; Jokinen et al., 2014, 2015) have concluded that in a-pinene ozonolysis, HOM formation from OH-initiated reactions is a minor channel, on the order of 10 %, compared to the ozone-initiated oxidation. As such, the impact of OH reactions on our results would be barely detectable. Nevertheless, we should have discussed OH oxidation in our manuscript, and we have now added segments to both describe that around half of the reacted a-pinene is due to OH reactions, and also explicitly stating (as we did above) why the role of OH is likely to be minor for the products (HOM) that are of interest in this work. For less oxidised products the importance of OH pathways will be nearly equal to the ozone-initiated pathways.

We also note that the molecules that were studied in detail in this work, in particular those given in Fig. 6, were compounds that earlier studies have shown to be formed specifically from ozone reactions, while the OH-derived HOM would normally have higher H-atom content and an odd number of oxygen atoms. Thus, our main analysis specifically focuses on known ozonolysis products. This also means that the potential multi-generation oxidation, i.e. where first-generation non-HOM oxidation products would react with OH to form HOM, would only to a minor extent be expected at these specific molecules. In addition, the likelihood of such multigeneration oxidation is quite low in our setup, since the flushout time is short, around 40 minutes, and the wall loss of condensable products shorter yet (on the order of a few minutes), and therefore OH reactions are going to be a relatively small sink for the oxidation products (Peräkylä et al., 2020; Bianchi et al., 2019).

Although a minor channel compared to ozone reactions, the main potential OH oxidation pathways that might form HOM are discussed in more details in Møller et al. (2020). Table 1 of that study shows the relevant intermediates from OH that are likely to produce HOM. The table includes one, albeit minor, channel ("1,5 allylic") that could form HOM where the D from C7 would be abstracted, so for a study focusing on products from OH oxidation, the importance of C7 might become larger than in our work.

Action: We have added discussion of OH chemistry in our manuscript section 4.3.

R2) Several referees stated that they would have liked to see more discussion of possible reaction mechanisms based on our findings so far. This is of course understandable, and we should have more clearly written why we decided to focus primarily on the use of the method itself and to give examples of the kind of information one could derive from selectively deuterated compounds. There were ultimately several reasons, with the main ones described below:

1. As also clear from many of the other referee comments, the experimental approach and the interpretation of the data are not self-evident, and we need to further improve the description of several aspects of the work. While the approach we deploy is very simple conceptually, i.e. exchanging selected H for D in the precursor molecules and observing the changes in the products, the detailed description of the resulting data (starting from Fig. 4) requires a considerable amount of explanation. We therefore felt it is better to write a paper with a clear focus on explaining the methodology and what kind of results can be achieved by using selectively deuterated precursors (thus we also chose a title saying "Selective deuteration as a tool for resolving..."). From our perspective, combining a highly detailed mechanistic analysis on top of the current findings would have made the paper very long and heavy.

2. Despite some of our mechanistic insights being quite clear cut, like D-loss becoming more likely at higher levels of oxidation and that C10 was the C-atom most often lost, any detailed mechanisms would still be highly speculative considering both the limited amount of deuterated precursors and the limitations of the experimental approach. For the latter, the relatively long timescales of our chamber experiment and our inability to utilise information about the radicals are the main limitations. For this reason, we opted for only broadly discussing the implications of our current results, and focusing on detailed mechanistic insights only when they appeared unambiguous.

3. Finally, given how promising this approach was found to be, more work is being put on synthesising additional selectively deuterated compounds, and we are aiming at acquiring more supporting data. The speculative mechanisms that we could provide based on the data presented in this manuscript might be verified or disproven by the additional data, and we therefore target another manuscript with a more mechanistic focus based on a more comprehensive data set.

Action: We have made it more clear in the manuscript why we chose the current focus in section 4.1., and clarified the experimental section. We also added a statement that a more mechanistically oriented study is planned in our conclusions.

R3 The referees would like more discussion on the kinetic isotope effect (KIE), as the seeming lack of a KIE was very surprising. This was indeed also a surprising result for us, given our earlier work in e.g. Rissanen et al. (2014). There

must of course be a KIE, even though it did not noticeably impact the HOM yields in our experiments. The key things here are the experimental timescales and the rate of competing reactions. In the work of Rissanen et al. (2014) using cyclohexene, the short timescale and full deuteration caused a massive effect, whereas in the current partial deuteration and long reaction time case the KIE does not appear to matter much, if at all, for the HOM formation. In other words, our conclusion is that the main HOM-forming pathways in this system have no closely competing pathways that would dramatically change the course of the reactions, under the conditions used here. In other words, the reactions had ample time to "run to completion" during the chamber residence time. In addition, it is also possible that there in many cases are several competitive pathways that ultimately lead to a similar result, and the partial deuteration only slows down one channel. We hope that our future work with additional precursor versions could shed more light on this interesting topic.

**Referee comments 1**

*In their article "Selective deuteration as a tool for resolving autoxidation mechanisms in α-pinene ozonolysis," the authors present an interesting effort to use deuterated isotopologues to understand the potential mechanisms for the formation of highly oxygenated molecules from oxidation of a common atmospherically important compound. Overall, it is a neat study that appears to be conducted carefully and knowledgeably. I recommend its publication after addressing mostly minor comments below.*

*Major comment:*

1. *I am left a little bit wanting for a more in depth discussion of mechanisms in the results section. There is a lot of great discussion of the data, but not effort to interpret the data in the context of existing (or new) proposed mechanisms other than that of Iyer et al. For example, on line 299, the pathway that removes D from the D3 is different from the proposed one, so it should account for some of the 50% not accounted for by Iyer, correct? Do you have any proposed mechanism for what that pathway is, and if so, does it account for any of the loss observed for the D2? If not, do you have any ideas about the loss of the cylclobutyl carbon? If those are all different pathways, it would almost fully explain the product. The tool the authors are using here seems quite powerful and it would be nice to see some new ideas for the mechanisms proposed or at least specifically discussed in the context of Figure 2. Can some of these formulas be given more clear example structures that would be consistent with existing (or new) mechanisms and account for the H/D data presented here?*

R4) We refer to our earlier response R2. The suggestions provided by the reviewer are exactly the type that we expect we will be able to address, if we can obtain supporting data from additional labelled compounds in the future. At this time, the system would be largely "underdetermined" and suggested mechanisms therefore highly speculative in most cases.

2. *In Figure 8 and its disucssion, why use number of oxygens as a proxy for lost? It seems to me that somehow breaking out the quantifaction by amount of D lost would be relevant. Part of the reason differences in yield are lower than in Rissanen is because not all of the ions have actually been impacted by a kinetic isotope effect. A figure like Figure 8,*

*but only for ions in which some portion are deuterated, or perhaps colored by mass fraction of deuterium. For example, if 50% of C10H14O7 is actually C10H13DO7, color it as maybe the weighted number of deuterium (0.5) or as fraction of hydrogens that are heavy (0.5\*0+0.5\*(1/14)). This might make an actually kinetic effect stand out. Otherwise, that implies there is no KIE - why might this be, given the results of Rissanen? What other reaction would outcompete? Given that there is usually going to be a non-deuterated abstraction available elsewhere in the molecule, shouldn't that usually outcompete? Could some of the differences in the relative yields and overall mass spectra be explained by KIE? For example, are the ions around 310 that are supressed in the D1 relative to the D0 (see Fig A2) heavily deuterated, thus possibly explained by KIE? Overall, it would be great to see a little more deep thought into what the mass spectra as a whole tell us about the KIE and also the potential mechanisms (comment 1)*

R5) Concerning the KIE, we refer to response R3. In addition, we would not conclude that "there is no KIE" even though no deuterium atoms were lost. The KIE might have caused another H-shift to take place instead of the D-shift. Nevertheless, we agree that it would have been very useful to plot Figure 8 coloured by fraction of D lost for each molecule. We have now made the plots where the colour scale shows the number of deuterium atoms lost on average per compound, and added the figure in the supplementary. Despite the modification, we do not see a clear effect from KIE. Nonetheless, the figure highlights the differing behaviours of the precursors, i.e. how $^7D_2$ HOMs mostly did not lose deuterium atoms, whereas $^3D_1$ and $^{10}D_3$ have (the colour scales have a larger range for the latter two precursors). Additionally, we can see that for $^3D_1$ precursor, most of the HOMs have lost on average some deuterium atoms (most points are in the green colour region), whereas for $^{10}D_3$ precursor HOMs, there are clearly some HOM that have lost more deuterium atoms than others (some points are yellow, where others are in the blue to green colour region).

Concerning the mechanistic determinations, we refer to response R2. Nevertheless, we thank the reviewer for pointing out the markedly lower signals at masses 308 ($C_{10}H_{14}O_7$) and 310 ($C_{10}H_{16}O_7$) Th in the spectrum for $^3D_1$. This can also be seen as the smallest orange fraction in Fig. 8a found for $^3D_1$. This result is indeed very intriguing when also noting from Fig. A3 that for $C_{10}H_{14,16}O_7$ there is a clear difference in fraction of D lost between $^3D_1$ and $^{10}D_3$, but at $C_{10}H_{14,16}O_9$, this difference is gone. Likewise, the yields of HOM with 8+ O-atoms (grey bars in Fig. 8a) are very close between the different labelled compounds. From a $RO_2$ radical perspective, it is clear that to form $C_{10}H_{15}O_{10}$, one necessarily has gone through $C_{10}H_{15}O_8$, and as such it is surprising that the products of the less oxidised radicals can decrease while the more oxidised products do not. It is possible that different structures of $C_{10}H_{15}O_8$ can form, and the ones that can undergo further H-shifts are not hampered by deuteration, while the isomers that cannot undergo H-shifts (and thus will terminate to e.g. $C_{10}H_{14}O_7$) are hampered. Based on the data we have, the latter option is supported, but without data on the radicals, also this finding remains highly speculative. If true, it would require us to rethink the way most people currently model the progression of $RO_2$ autoxidation, but we are hesitant to make such a claim based on the data at hand. Regardless, we have added discussion on this matter to the text.

*Technical comments:*

L22 →: *Defining "HOM" as the plural "highly oxygenated organic molecules" leads to prevelant odd grammar. In some places, it is fine, such as line 24 ("HOM ... have been shown"). However, it means it is basically grammatically impossible to refer to a single highly oxygneated organic molecule. For example, line 87, "a closed shell HOM" means "a closed shell highly oxygneated organic molecules", which is not grammatically correct, and there isn't really a good way to make this sentence grammatically correct if HOM is defined as plural. I recommend defining HOM singularly and using HOMs as the plural.*

R6) We revised the definition of HOM to being singular to enable grammatically correct use.

L34: *Run-on sentence. Instead of "[and] it can be a", use "as in the case of"*

L39-40: *I'm not sure I understand what this sentence means, try to clarify a bit what was shown in that paper.*

L48 *& 111: Delete "molecules"*

L183-185: *Run-on sentence*

L233 & 236: *Should be "by a large margin", not "with a large margin"*

R7) L34-L233&236 are corrected to text. We added clarification to L39-40: "They also showed that D-atoms were exchanged to H-atoms in contact with water vapour in cases where the initial C-D bond was broken, and the D became attached to an oxygen through an O-D bond."

L238: *Do I understand correctly that it is not completely inactive, just mostly so? Or can even an inactive carbon lose some of its H/D?*

R8) We agree with the reviewer that the terminology was not clear enough in our manuscript. The terms "active" and "inactive" are far from perfect, as they can be somewhat ambiguous. However, we decided to use them because our data are also not perfect due to various uncertainties. We have now made the definition of "active" and "inactive" more clear, saying that "active" means that we observed D-abstractions taking place at fractions above our uncertainty margins. This simpler definition does not take into account the potential effect that we were trying to express on line 238, namely that it is possible that a specific C-atom might normally be active, but the deuteration itself makes it inactive. Nevertheless, we did not see indications of this type of behaviour in our data, and we feel that this new definition is less ambiguous than the earlier one.

L266-267: *I'm not sure I understand what the authors mean by there being "numerous isoprenes for every molecule". Do they mean for each molecule they will investigate multiple D/H isotopes? Or do they mean something about the natural distribution of isotopes? Re-word.*

R9) Added "numerous D/H isotopes" for clarity.

L326: *What was the value reported by Bianchi et al.?*

R10) Added the values (which ranged from 1.2 (+1.2/-0.72) to 7.0 $\pm$ 3.5 %) reported by Bianchi et al. (2019) to text.

- *Methods section is written very informally. I'm not actually sure this is an issue, but it does feel a little odd. For example "the PTR instruments were calibrated for sensitivity, and we used that information to get the precursor concentrations from the ion signal" - the use of the word "get" here is a very informal use. A more typical writing style for a manuscript might be "the PTR instruments were calibrated to estimate a response factor, which was used to convert ion signals into precursor concentrations." The informal language is seen throughout the manuscript, but particularly in the methods. I think whether this is an issue or not is an editorial decision.*

R11) We aimed to use mostly active tense while writing, and this may be part of the reason why the reviewer finds the text to be informal. In addition, we agree that the wordings used in some sentences could be improved. In the example by the reviewer, our suggestion is to replace the word "get" by "determine" in addition to other rewording: "we conducted sensitivity calibrations to the PTR instruments to obtain calibration factors, which we used to determine the precursor concentrations from the ion signal". Additionally, we did some other small corrections to improve the formality of the whole text.

- *Figure 1 is very helpful and clear*

- *Figure 4 - I am generally strongly opposed to bar charts in log space, because the idea of a bar chart is that height or area of each bar should represent relative magnitude, which is not true in log space because there is no true zero and the relative heights of bars provide little information about their relative magnitude. However, given that this particular is illustrative and not particularly focused on qunatitation, I'm not necessarily opposed to it here.*

R12) We agree with the reviewer that bar plots in log scale should typically be avoided. However, in this case, we are plotting mass spectra, which are almost always plotted as bar charts, hence we feel that the usage should not be misleading.

- *Figure A2 - label left and right as monomer and dimer. It is also very interesting that the overall distributions of ions is not the same for each precursor (for instance ratio of cluster around 310 and 315 is different for the D1 and D3 precursor), do the authors have any thoughts on this?*

R13) We thank the reviewer for this suggestion on the labelling, as it makes the plot easier to read. Concerning the peak ratios, we refer to response R5.

**Referee comments 2**

*Review of "Selective deuteration as a tool for resolving autoxidation mechanisms in $\alpha$-pinene ozonolysis" by Meder et al., (egusphere-2022-1131)*

*In this manuscript, Meder et al. investigated ozonolysis of normal and partially deuterated $\alpha$-pinene, focusing on the distribution of highly oxygenated molecules (HOMs) with different numbers of D atoms and HOM yields when different partially*

*deuterated were used. This technique of utilizing partially deuterated precursors indeed will help to better understand reaction pathways, but the authors should have gone much further beyond their current discussion. Especially, they are advised to connect their observation to potential mechanisms, and at least try to suggest the formation routes of some of the most abundant products. I would recommend major revision of this manuscript.*

185     *Main concerns,*

1. *The authors seem to overlook OH radical formation in the ozonolysis of $\alpha$-pinene. In fact, the OH yield is pretty high, which leads to a dispensable fraction of OH+$\alpha$-pinene in this system. I mean, then the observed distribution of (partially deuterated) HOMs does not necessarily reflect the ozone chemistry. Also, OH might react with the HOM products, further altering their distribution. How to evaluate the impact of secondary reactions on HOM distribution? By the way, why*

190     *wouldn't the authors give the extent of consumption of precursors in each case?*

R14)   We refer to our response R1 concerning the impact of OH on the HOM formation. Concerning the suggested loss of HOM by OH, the wall loss rate ($\approx 10^{-2}$ s$^{-1}$) of HOM in our chamber far outruns the loss via reactions with OH ($< 10^{-4}$ s$^{-1}$).

Concerning the question about reporting precursor consumption in each experiment, we realise that we should have

195     more clearly described our experiment setup in the methods section 3.1.1. We mention that the chamber was a Continuous Stirred-Tank Reactor (CSTR) but did not describe explicitly that this meant that we ran it as a "steady-state chamber". This means that we continuously fed precursors into the chamber throughout the experiments, aiming for a stable concentration resulting from a balance between sources (inflow) and sinks (outflow, chemical reactions, and wall losses). As such, the absolute precursor consumption varied with the amount of precursor that was fed in, which often varied even

200     across one experiment, and therefore reporting the consumed precursor concentration is much less relevant than in typical batch mode chamber experiments. We have now added more details on this to the experimental set up section 3.1. We also added the estimate for precursor consumption based on the relative ratio of chemical reactions vs outflow. With an ozone concentration of $43 ppb = 1.0578 \cdot 10^{12} cm^{-3}$ and an AP+O3 reaction rate coefficient of $k_1 = 9.06 \cdot 10^{-17} cm^3 s^{-1}$, the loss rate of AP is $9.58 \cdot 10^{-5} s^{-1}$. Over an average residence time of 40 min in the chamber, this means that roughly

205     $40 min \times 9.58 \cdot 10^{-5} s^{-}1 = 0.230 = 23\%$ of the injected AP reacted before being flushed out from the chamber.

2. *It was stated that "autoxidation was perturbed in predictable ways by the deuteration" (Line 40, Page 2). On the other hand, there is only one paragraph that discussed confirmation of a detailed reaction pathway by Iyer et al. 2021. The rest just ends up with "C-3 and C-10 are active and C-7 is mostly inactive". If they can exclude potential interferences from OH+$\alpha$-pinene and OH+HOMs as I raised above, the authors might be able to figure out routes beyond the Iyer et*

210     *al. 2021 study.*

R15)   The quote from Line 40 relates to the Rissanen et al. (2014) study on cyclohexene, and concerns the impact deuteration had on their results. We are not sure how this relates to our study in this context, but concerning the mechanistic details we refer to our response R2, and concerning the role of OH, we refer to R1.

3. *Figure 6 is confusing to me.*

    – *Please at least give one example of the definition in the caption. For example, in the case of C10H14O7, when 3D1 was the precursor, -0D corresponds to C10H13DO7 and -1D corresponds to C10H14O7? When 7 D2 was the precursor, -0D corresponds to C10H12D2O7, -1D corresponds to C10H13DO7, and -2D corresponds to C10H14O7?*

R15a) The interpretation of the reviewer is correct. We added an example to the figure caption to clarify the figure.

    – *Why aren't there a -3D in the upper panel of Figure 6? Do the authors suggest loss of 3D in 3D1 was not observed?*

R15b) The reviewer is again correct, we never observed loss of all 3 D-atoms, and therefore this legend entry was not included. We added also this to the caption for clarity.

    – *The color scale in Figure 6 might be modified for a better differentiation.*

R15c) We are not sure which part of the colour scale is hard to differentiate. We tried to choose a scheme that was clear both in colour and for black-and-white by going from lighter to darker colours. We also include hashing for this purpose. Regardless, we have now made the hashing more pronounced, hoping that this removes any ambiguity between the colours, and changed the colour map to better match the aesthetics of the other figures.

*Minor comments*

1. *(Paragraph 2, Page 2), selective deuteration has been used to study autooxidation mechanisms in the 1,3,5-trimethylbenzene + OH system (Wang et al., ACP 2020, 20, 9563-9579), which is highly relevant to this study.*

R16) We thank the reviewer for making us aware of this study, as we had not noticed this use of selective deuteration for HOM studies before. We now mention this study in the introduction. We also note that the referenced study is a good example of how the deuteration could be used to draw some conclusions on the mechanisms, but as the authors also conclude, the suggested mechanisms remains speculative.

2. *The author states that "the purity of each compound was >95%" in line 66, and that "PTR measurements suggests that over 88% of the measured pinene in our chamber contained exactly the number of D-atoms specified" in Line 218. The two numbers are close, but could still lead to a problem for the purpose of a mechanism study. Please revise or justify.*

R17) These are two independently determined values, and as such we cannot revise them in any way. It is also to be expected that the purity of a compound decreases by the time it is actually sampled from the chamber compared to the initial synthesis for various reasons, including instrumental limitations and experimental impurities. Furthermore, we report this 88 % because it provides another estimate of our uncertainty, namely that if D-loss is observed at a level around 10 % or less, we cannot conclude that this deuterated C-atom would have been "active".

3. *"Inactive" is not a perfect word to describe whether or not a D atom goes through autooxidation. Since this manuscript judge by the distribution of products, but even for the "inactive" precursor with selective deuteration at the C-7, products with loss of D atoms were observed, which suggests that something happened to the C-D bond.*

245    R18) We refer to our responses R8 and R17.

4. *(Line 302), one can also argue that C19H28O11 can be formed from a 3D1 precursor, since the signal of C19H28O11 in the case of a 3D1 precursor could be attributed to mostly loss of a D atom. Does this mean that C19H28O11 is formed via one monomer that reacts at C-3 and the other at C-10.*

R19) If we interpret the reviewer's comment correctly, they are suggesting that also C-3 might be lost during the formation of

250       the C19H28O11 dimer. This should not be the case, as $^{10}D_3$ results indicate that the lost C-atom is almost always C-10. Otherwise, it would be very hard to explain the abundant loss of D-atoms in the $^{10}D_3$ case. The loss of D from $^3D_1$ can be explained by a D having been abstracted in the majority of cases when forming this dimer. However, it is indeed a very viable formation pathway for this dimer, that one monomer loses the C-10, while the other (or the first) monomer often undergoes an abstraction from C-3.

255 **Referee comments 3**

*Major Comment*

– *The first thing I would highlight is the effort gone into presenting this work in a digestible way. The colour coding of the deuteration, and then presenting the reaction routes with this colour coding makes working out how the products are potentially formed straightforward. The big picture results of this study is that vinylic C-3 and allylic methyl C-10*

260       *carbons are active but the cyclobutyl ring carbon, C-7 is not. Via the C-3 isomer, 50% of its loss is consistent with the literature Iyer et al. mechanism. In general, only one D loss occurs in the oxidation process, but there is a mechanism where the CH3 is lost, and strong evidence for this is seen via the 10D3 isomer data.*

– *Regarding the total HOM yield, there is little difference between the deuterated isomer (and normal alpha-pinene) and is around the 5% yield; but the 7D2, which is inactive, does have the highest yield. However, the lack of significant kinetic*

265       *isotope for the active 3D1 and 10D3 is surprising. There probably should be a little bit more about potential mechanisms that produce little or no kinetic isotope effect. Is there a predicted isotope effect via the Iyer et al. mechanism? However, things are probably simplified too much, and while I like this approach there should be acknowledgement of the extra stuff going on. The most obvious is that at the early stage of ozonolysis a significant amount of OH is made, and this is going to react with the pinene to make different peroxy radicals that are going to undergo auto-oxidation leading to*

270       *HOM formation. You note that for 3D1 via the Iyer mechanism accounts for about 50% of the signal. Could it be that OH chemistry is accounting for this missing signal?*

R20) We thank the reviewer for highlighting the effort we have put into presenting the work. For the discussion on KIE, we refer to our response R1.

– *While I might be showing my ignorance of the experimental details, as I understand it a slow flow of O3/alpha-pinene*

275       *is introduced into the chamber and it typically takes about 40 minute to flow out. So the experiment is essential a snap*

*shot in time at 40 minutes. However, the system never seems to reach this 40 minute snap shot in time, and the products are evolving over a much longer timescale. Does this indicate that the surface of the chamber is playing a role in these experiments? If this is the case, can you be sure that the products are the result of only gas-phase chemistry. I think some acknowledgement of the role of the chamber surface is required. Or can it be dismissed?*

R21) We refer to our response R14. We indeed did not describe the chamber operation in enough detail, and the presented results are not a snapshot at 40 min, but rather consists of a range of different oxidation time scales, with an average of 40 min. We have now made this clear in section 3.1. The chamber surfaces act as the main sink for the formed HOM, but should otherwise not considerably perturb the spectra.

– *Overall, this work has demonstrated that via deuteration of the target alkene some of the HOMs that form via ozonolysis can assigned to specific isomerization steps. This is useful information. There are a number of things that can be done to put this work into better perspective, but overall I have no problem recommending this paper for publication.*

*Specific comments*

L80 *The O8-RO2 product remains the only experimentally and computationally supported HOM-forming pathway in this system. This mechanism involves the 3D1 bond breaking. Then this route should show a kinetic isotope effect? Is there a number from theory?*

R22) We did not calculate how much slower this H-shift would become in the case of a D-shift, but Iyer et al. (2021) report that the H-shift rate is $2 s^{-1}$. When considering that deuteration typically slows down the reaction by factor 20-100, this still leaves rates of $0.02 - 0.1 s^{-1}$, which are well within the time scale of our experiment. Given the low concentration of reactants, the $RO_2$ lifetime with respect to bimolecular reactions are also expected to be long enough for these reactions to take place. As such, the KIE would be there, but it would simply not impact the product yields very dramatically under our experimental conditions. We refer to our response R3 also.

L130 *if the H-shift is 1000 times faster than any competing reaction, substituting for a D-shift will barely impact the branching of the reaction, While true about the branching ratio, but it will take longer to form if deuterated, hence yields should be lower, certainly at early times.*

R23) We refer to our two responses R21 and R22 above.

L145 *"We injected $\alpha$-pinene into the chamber using one of two methods: an overflow set-up for small amounts of precursor, and a syringe pump set-up when there was enough precursor for using a 5 μl syringe to take a sample. In both cases, the evaporated α-pinene was injected into the chamber with a small N2 flow." So what was the typical [pinene] in the chamber at the start? I can see the answer to this in Figure 5. Perhaps it should be stated in the experimental section.*

R24) We refer to our responses R14 and R21. As we ran in steady-state mode, the concept of "at start" is not really valid.

*L117* *"We excluded all isotopes that contained deuterium, because using selectively deuterated precursors distorts these signals." Is this exclusion only for determining the instruments sensitivity?*

R25) Yes, the referee is correct. We clarified the section in the text.

*L199* *"We calculated the HOM yield from the production and loss terms of HOM when the concentration of HOM [HOM] did not change significantly over time (Ehn et al., 2014)": But your experiments have a constant residence time in the chamber, so what you means by "time"*

R26) We refer to our responses R14 and R21.

*L223* *"Similarly HOM dimers can contain between twice the number of deuterium atoms in the precursor and zero deuterium atoms." There needs to be more on HOM dimers. Perhaps introduce them in section 2.2. What is a typical HOM dimer yield in this study? Are they at steady-state?*

R27) We added clarification on what is a HOM dimer in the scope of this study to sections 2.2 and 3.2. We refer to $RO_2 + RO_2 \rightarrow ROOR$ accretion products as HOM dimers following the commonly used nomenclature. The total dimer yields can be seen in Fig. 8a (blue bars) for each precursor, however, the high uncertainties of the values should be noted. The values range between 0.4 and 0.8 %. As for the steady-state, we refer to our response R14.

*L242* *"As can be seen from Fig. 5, optimal steady-state conditions were not achieved in all cases, in particular when the injection was performed using the overflow setup (Fig. A1a), as was the case for 7D2 and 10D3." Again, as the time in the chamber of constant, i.e. time before measurement, where is this change with time coming from? I presume it is linked to things going to the walls before the walls are at steady-state, which is not the same as the steady-state of equation (2). If you run a model of the system, what time does this predict for steady-state, where I presume the model will have loss to the wall as a constant.*

R28) We refer to our response R14.

*L263* *"as in normal experiments the radicals and closed-shell species would be at different integer masses and thus easily distinguishable." But you have done "normal experiments" when doing D0. So can you show some radical data? Are the radicals at steady-state?*

R29) It is true that distinguishing radicals in the $D_0$ case is more straightforward compared to the other isotopes, but as the visible radical signals were primarily just $C_{10}H_{15}O_{8,10}$, in accordance with earlier studies, we did not find it useful to show these explicitly. The radicals of this system are generally well-known (e.g. Berndt et al., 2018). Concerning the steady-state, radical lifetimes are on the order of minutes or less in our experiments, which is a short time compared to precursor concentration changes, and as such, we would say that the radical were almost always very close to a steady state.

L299 *"For the 10D3 C10H14O7, about 20 % has lost a D-atom, while the other 80 % behaved according to expectations from Iyer et al. (2021)." 80% if no other mechanisms is considered!*

R30) We assume the reviewer points out the possibility of misleading the reader with this choice of word. We have modified the part to point out the possibility of other pathways. "For the $^{10}D_3$ C10H14O7, about 20 % has lost a D-atom, while the other 80 % behaved according to expectations from Iyer et al. (2021). However, the Iyer et al. (2021) mechanism is not necessarily the only mechanism through which all of the 80 % of the signal is formed, as there can be other mechanisms that could explain parts of the signal."

L338 *"However, this change can be compared to the change of two orders of magnitude observed by Rissanen et al. (2014) for fully deuterated vs non-deuterated cyclohexene, suggesting that deuteration had a minimal role, if any, for the HOM yields of our precursors." It is clear that C—D bonds have been broken in this study, but to observe slight change in the yield is surprising. I think a little more speculation on this observation is required.*

R31) We refer to our response R3.

L350 *"This may be an indication that the D-shifts were still fast enough to outcompete other reaction pathways despite the deuteration. On the other hand, it is possible that the autoxidation could proceed through the next most competitive pathway not shut down by the deuteration and end up losing deuterium atoms later in the process, especially in the case of the more oxidised products." While deuteration might not change reaction paths, it should slow the rate to products? So the second sentence explanation is more likely. Would you like to speculate how the D is happening later in the process and not show a significant kinetic isotope effect?*

R32) We refer to our response R3.

**Referee comments 4**

*Meder et al. discuss the use of deuterated α-pinene standards to investigate its oxidation by ozonolysis. The authors selectively replaced hydrogen atoms with deuterium atoms to investigate at what carbon centers hydrogen/deuterium abstraction occurred. The authors used a high resolution chemical ionization orbitrap mass spectrometer to differentiate peaks associated with the deuterated samples and based on obtained mass spectra, discussed likely oxidation mechanisms. The work presented in this manuscript is novel and attempts to address a challenging gap in knowledge regarding oxidation mechanisms for α-pinene ozonolysis. Overall, the manuscript presents important data that should be accepted for publication in ACP once some major comments have been addressed.*

    *Major comments*

– *Although this manuscript presents some convincing arguments, I feel it would benefit from some structural changes that would significantly streamline and clarify the work presented. Primarily the manuscript should be reorganized to follow a more logical flow.*

1. *For example, the "selective deuteration and autoxidation sections" contains a mixture of literature motivation and methods used, and as such could be instead integrated into the "introduction" as well as the "methods" sections as is appropriate.*

R33) The reviewer is correct, section 2 does contain a mixture of things that might normally be explained in the introduction or the methods. We did consider this approach as well, but we felt the current division had more benefits than drawbacks. In particular, the description of autoxidation chemistry is very relevant for understanding the work we did, but it would become the main focus of the introduction if it was included there. Likewise, the concept of using selective deuteration could have been given in the introduction, but it would also have made it very long. Now, with this section following the description of the deuterated compounds actually used in this work (which often is included in the Methods), we can much more concretely describe how the different pathways would be impacted by the selective deuterations, as in Fig. 2. For these reasons we prefer to keep section 2 in its current format. We did reorganise the Methods section to follow a more logical flow.

2. *Additionally, the authors should consider reorganizing the structure of their results and discussion section (e.g., switching the order of sections 4.1 and 4.2). Data should be presented first followed by in its interpretation and discussion, i.e., there should not be a section on "interpreting the mass spectra" before "experiment overview" where mass spectra should be introduced.*

R34) We agree with the reviewer and changed the order of sections 4.1 and 4.2.

– *This manuscript would also benefit from some additional details regarding experimental conditions, instrument details and uncertainties, etc.*

1. *Although the authors state that they were focused on the interpretation of general trends in the mass spectral data between different deuterated species, including these quantitative details would help readers to interpret the data presented and importantly the limitations of the work that are described by the authors.*

R35) We in part refer to R14, as we indeed did not describe the experiment setup properly, which caused a lot of confusion about our data. With this made clearer, we feel that the most important parameters are visible in Fig. 5. Having a table of experimental conditions is less useful in this study than in most other chamber studies, as the a-pinene concentration varied throughout the experiments.

2. *Furthermore, a lot of useful information is present in the manuscript, but it can be challenging to find in its current organizational state. For example, some of these important details (e.g., $\alpha$-pinene mixing ratios) were not stated in the text, but instead had to be determined via figures.*

R36) See our response above.

3. *Additionally, even though the authors reference existing literature for the instrumental methods, enough details should be given so that readers know the essentials without having to refer to other publications.*

R36) We agree with the referee, and have now extended the instrument description section and added more to the description of the experimental set up in addition to reorganising the sections.

– *The number of experiments conducted is unclear. Were replicates of these experiments conducted or just the 4 listed? Given the large experimental uncertainty, replicate measurements would be of significant benefit to help constrain these results and allow for a more rigorous interpretation of the experimental data. Furthermore, a summary of experiments and conditions would be beneficial.*

R37) The referee is correct concerning the benefits of replicates, but unfortunately, due to the very limited amount of deuterated precursors, we could only run the experiments once per precursor. This adds to the already high uncertainties from e.g. the CI-orbitrap. This was discussed in the text, but will be clarified further in the experiment overview section. We also refer to our response R2.

*Minor comments*

– *The color scheme of the figures could be improved. It would be very beneficial for the authors to stick with a common color scheme and employ this throughout the manuscript. Also, shaded regions and lines do not have to have a pattern if an appropriate color scheme is chosen.*

R38) We are not sure in what way the colour schemes should be improved, i.e. whether the reviewer found the colours hard to discern, or if the case was more concerning aesthetics. The patterns of shaded regions were included specifically to make them distinguishable for as many people as possible. As concerns the common colour scheme, we believe we do use the same colors consistently when referring to the same parameters in different figures. Regardless, we have modified the figures as described in our response R15c.

– *The authors are attempting to determine the mechanism through which $\alpha$-pinene undergoes ozonolysis. However, OH is also generated since an OH scavenger was not used during the experiments. The authors do not comment on this in the manuscript discussion. Further mention of this should be discussed. In particular, do the authors expect that the kinetics of OD vs OH generated to impact potential OH oxidation chemistry?*

R39) We refer to our response R1.

– *There is a lot of discussion in the introduction regarding decrease in reaction rates due to the kinetic isotope effect (KIE), yet the authors claim that a KIE is not observed with their data. Why do the authors propose this is?*

R40) We refer to our response R3.

– *The authors state that the data were divided into two periods "Chigh" and "Clow" when the reaction rates for k[$\alpha$-pinene][O3] were 0.5 ppt s-1 and 0.015 ppt s-1 Can the authors please elaborate on how these periods were chosen?*

R41) As described in the manuscript, we had very little of some of the samples, and it was hardly guaranteed that we would get enough signal for our experiments in all cases. Therefore, our aim was merely to get two different concentrations that could be reached for each compound.

– *If the chamber is being operated in a steady state continuous flow mode, why is the signal for the precursor dropping over time but ozone is relatively stable?*

R42) The precursor concentration was dropping due to decreased evaporation as the sample was running out. The ozone stayed largely constant despite the change in precursor VOC because the low VOC concentration was only a minor sink for ozone compared to the flushout rate. We refer also to our response R14.

– *The authors cite Rissanen et al. stating that "D-atoms can exchanged to H-atoms in contact with water vapour in cases where a C-D bond was broken". The experiments were conducted at RH<1% so it is unlikely that these processes should be significant under the experimental conditions. However, under ambient conditions RH is much higher. How do the authors expect RH might affect the experimentally observed results?*

R43) Even at RH=1%, there will be around $10^4 - 10^5$ collisions per second with $H_2O$, and therefore we have found that all D-atoms attached to O-atoms efficiently exchange to H-atoms in our chamber even without actively adding H2O. We have also tested adding H2O, and the results did not change, meaning that the $D \rightarrow H$ exchange was complete also at low RH. The RH itself has also not been seen to impact the autoxidation or HOM formation (see e.g. Li et al., 2019).

*Specific comments*

L66 *The authors state "The purity of each compound was >95% as determined by 1HNMR spectroscopy and we were unable to observe residual proton resonances associated with the deuterated carbon positions for any sample..." This is misleading as only NMR spectra are given for the 3D1 α-pinene sample. Was a purity of >95% also obtained and characterized by NMR for the other two deuterated α-pinene samples synthesized?*

R44) The referee is correct that only the $^1$HNMR spectra for the $^3D_1$ precursor are given in the supplementary, and that the same purity was obtained for the other two precursors. However, the latter information can be found from the cited studies (i.e. Upshur et al., 2016, 2019). Nevertheless, we have now added the $^1$HNMR spectra for the other two deuterated precursors in the supplementary.

L137 *"RH" should be in brackets.*

R45) We have added the brackets.

L150 *How much did these concentrations vary? Please state a range in the text. These can be determined from Figure 5 but should be stated in the text.*

R46) The precursor concentrations are now added to the text in section 4.1. The $C_{low}$ sample corresponds to precursor concentration ranges of $1.3-1.7$ ppb, $0.6-1.0$ ppb, and $1.4-1.8$ ppb for the $D_0$, $^3D_1$, and $^7D_2$ precursors, respectively. The $C_{high}$ sample corresponds to $7.6-8.0$ ppb, $4.6-6.0$ ppb, and $5.0-5.4$ ppb for the $^3D_1$, $^7D_2$, $^{10}D_3$ precursors.

460  L157 - 158 The authors state "the instrument has been shown to be effective in detecting HOMs". It would be beneficial for the authors to more information here regarding instrument sensitivity, etc. Perhaps adding parts of the "data analysis" section here would be helpful.

R47) We added some details to the instrument description portion (section 3.1.).

L162 The authors state "the results were comparable" in reference to the using both a VOCUS-PTR-ToF and PTR-ToF to
465  conduct experiments. Without showing the data or giving any sort of quantitative comparison, it is hard to make these statements. Consider rephrasing or adding details to convince the readers that this is in fact the case.

R48) We agree with the referee, the sentence was poorly formulated and we decided to remove it. Both instruments were calibrated in the same way before use.

L173 Please state the sensitivity here.

470  R49) We have added the sensitivities or the calibration factors in the text as suggested. The calibration factors were $15$ cps/ppb for the PTR-ToF used for $D_0$, $^7D_2$, and $^{10}D_3$ precursor data, and $180$ cps/ppb for the Vocus-PTR used for the $^3D_1$ precursor data.

L177 The authors state "We excluded all isotopes that contained deuterium, because using selectively deuterated precursors distorts these signals". It is unclear what the authors mean here. If you are trying to measure deuterated samples, then
475  understanding the instruments response to these species is key to obtaining quantitative data.

R50) The referee is correct, the sensitivity towards deuterated samples is essential. However, in this section the aim was to utilise known natural isotope abundancies to assess the sensitivity as a function of signal intensity, but due to the active deuteration of our samples, we could no longer assume a natural isotope ratio. For this reason we only used C, N, and O isotopes. More generally, in this case the composition of an ion is irrelevant, only the signal strength matters. We have
480  added clarifications to the text.

L214 Please state the limit of detection of the instrument.

R51) We have added the limit of detection of the instrument to the text. Estimating from the instrument's sensitivity behaviour, we found it to be $10^{-6}$cps/cps (normalised units of signal intensity) or $10^{-4}$molecules $\cdot$ cm$^{-3}$ with a large uncertainty of at least $-67\%/+200\%$ (see text for the discussion of the uncertainty).

485  L240 This section is labeled "experiment overview". Data are being discussed here and not the experimental procedure. Consider renaming this section.

R52) We have modified the section title to "Overview of experiments".

*L241 The authors state "we conducted four experiments with..." Were these 4 different types of experiments or only 4 experiments total? It is unclear if or how many replicates of these experiments were conducted.*

490  R53) We conducted four experiments in total, we have clarified this in the text.

*L308-311 It would be helpful to include the fraction of times that e.g., Ds were lost. Statements such as "often" or "rarely" are somewhat ambiguous.*

R54) We agree with the reviewer on the ambiguity of the used words. However, these are intentionally used since the data has large uncertainties and we are not aiming to give very specific results. We refer to our response R8 (we have added a
495  more specific definition in the text).

*L337 The authors state "...the differences in steady-state precursor concentrations between the experiments". There are not explicitly given in the manuscript (other than what can be deduced from Figure 5) and should be added as part of the methods section. Also, the term steady state should not be used here as it implies that the concentrations are not changing. However, according to Figure 5, and the authors themselves on Line 242, this is not true for several of the*
500  *experiments.*

R55) We have added the concentrations that correspond to the $C_{low}$ and $C_{high}$ regions, see our response R46. We agree with the reviewer on the use of "steady-state" and have removed the word, however, considering HOM chemistry, the conditions are extremely close to a steady state.

*Fig8 Please state slopes and offsets for the linear fits listed in graphs 8b-d.*

505  R56) We have modified the graphs 8b-d to better show what we wanted. The linear fits are not important, but the line is there to guide the eye acting as a trendline and make the comparison to the 1:1 line easier. With the new naming, we don't find it important to state the slopes and offsets of the lines as they are irrelevant.

*L403 "there was a bug in the pump" is unclear. Was this a mechanical issue? Can the authors please clarify this in the text.*

R57) It was a communication problem between the software that automatically controls the orbitrap and syringe pump and it
510  resulted in the syringe pump pumping at a much greater rate than what was shown on the syringe pump screen or set in the controlling software. We mention this as it explains some of the large background in our experiments, but it is not otherwise relevant. We have reworded the part to make it clearer.

*FigA2 Were the $\alpha$-pinene oxidation products not measured by orbitrap? As such their mass spectra should not be in "unit mass resolution". Also, it would be of benefit to either label the peaks or list the m/z associated with each HOM for ease of*
515  *reading.*

R58) The signals are shown in unit mass resolution for simplicity. For example, in this way they y-axis unit can easily be given in cps, whereas plotting a distribution would cause ambiguity in the y-axis values. In addition, the spectra would look almost identical also with HR with this broad mass ranges plotted, so we do not see the usefulness of plotting the data in another way. We agree with the referee that labelling the peaks would make the figure easier to read, and have modified the figure accordingly.

**References**

Berndt, T., Mender, B., Scholz, W., Fischer, L., Herrmann, H., Kulmala, M., and Hansel, A.: Accretion Product Formation from Ozonolysis and OH Radical Reaction of alpha-Pinene: Mechanistic Insight and the Influence of Isoprene and Ethylene, ENVIRONMENTAL SCIENCE & TECHNOLOGY, 52, 11 069–11 077, https://doi.org/10.1021/acs.est.8b02210, 2018.

525 Bianchi, F., Kurtén, T., Riva, M., Mohr, C., Rissanen, M. P., Roldin, P., Berndt, T., Crounse, J. D., Wennberg, P. O., Mentel, T. F., Wildt, J., Junninen, H., Jokinen, T., Kulmala, M., Worsnop, D. R., Thornton, J. A., Donahue, N., Kjaergaard, H. G., and Ehn, M.: Highly Oxygenated Organic Molecules (HOM) from Gas-Phase Autoxidation Involving Organic Peroxy Radicals: A Key Contributor to Atmospheric Aerosol, Chemical Reviews, 119, 3472-3509, https://doi.org/10.1021/acs.chemrev.8b00395, 2019.

Ehn, M., Thornton, J. A., Kleist, E., Sipilä, M., Junninen, H., Pullinen, I., Springer, M., Rubach, F., Tillmann, R., Lee, B., Lopez-Hilfiker, F.,
530 Andres, S., Acir, I.-H., Rissanen, M., Jokinen, T., Schobesberger, S., Kangasluoma, J., Kontkanen, J., Nieminen, T., Kurtén, T., Nielsen, L. B., Jørgensen, S., Kjaergaard, H. G., Canagaratna, M., Dal Maso, M., Berndt, T., Petäjä, T., Wahner, A., Kerminen, V.-M., Kulmala, M., Worsnop, D. R., Wildt, J., and Mentel, T. F.: A large source of low-volatility secondary organic aerosol, Nature, 506, 476–496, https://doi.org/10.1038/nature13032, 2014.

Iyer, S., Rissanen, M. P., Valiev, R., Barua, S., Krechmer, J. E., Thornton, J., Ehn, M., and Kurtén, T.: Molecular mechanism for rapid autoxidation in $\alpha$-pinene ozonolysis, Nature Communications, 12, 878, https://doi.org/https://doi.org/10.1038/s41467-021-21172-w, 2021.
535

Jokinen, T., Sipilä, M., Richters, S., Kerminen, V.-M., Paasonen, P., Stratmann, F., Worsnop, D., Kulmala, M., Ehn, M., Herrmann, H., and Berndt, T.: Rapid Autoxidation Forms Highly Oxidized RO2 Radicals in the Atmosphere, Angewandte Chemie International Edition, 53, 14 596–14 600, https://doi.org/https://doi.org/10.1002/anie.201408566, 2014.

Jokinen, T., Berndt, T., Makkonen, R., Kerminen, V.-M., Junninen, H., Paasonen, P., Stratmann, F., Herrmann, H., Guenther, A. B.,
540 Worsnop, D. R., Kulmala, M., Ehn, M., and Sipilä, M.: Production of extremely low volatile organic compounds from biogenic emissions: Measured yields and atmospheric implications, Proceedings of the National Academy of Sciences, 112, 7123–7128, https://doi.org/10.1073/pnas.1423977112, 2015.

Li, X., Chee, S., Hao, J., Abbatt, J. P. D., Jiang, J., and Smith, J. N.: Relative humidity effect on the formation of highly oxidized molecules and new particles during monoterpene oxidation, Atmospheric Chemistry and Physics, 19, 1555–1570, https://doi.org/10.5194/acp-19-
545 1555-2019, 2019.

Møller, K. H., Otkjær, R. V., Chen, J., and Kjaergaard, H. G.: Double Bonds Are Key to Fast Unimolecular Reactivity in First-Generation Monoterpene Hydroxy Peroxy Radicals, The Journal of Physical Chemistry A, 124, 2885–2896, https://doi.org/10.1021/acs.jpca.0c01079, pMID: 32196338, 2020.

Peräkylä, O., Riva, M., Heikkinen, L., Quéléver, L., Roldin, P., and Ehn, M.: Experimental investigation into the volatilities of highly
550 oxygenated organic molecules (HOMs), Atmospheric Chemistry and Physics, 20, 649–669, https://doi.org/10.5194/acp-20-649-2020, 2020.

Rissanen, M. P., Kurtén, T., Sipilä, M., Thornton, J. A., Kangasluoma, J., Sarnela, N., Junninen, H., Jørgensen, S., Schallhart, S., Kajos, M. K., Taipale, R., Springer, M., Mentel, T. F., Ruuskanen, T., Petäjä, T., Worsnop, D. R., Kjaergaard, H. G., and Ehn, M.: The Formation of Highly Oxidized Multifunctional Products in the Ozonolysis of Cyclohexene, Journal of the American Chemical Society, 136, 15 596–
555 15 606, https://doi.org/10.1021/ja507146s, pMID: 25283472, 2014.

Upshur, M. A., Chase, H. M., Strick, B. F., Ebben, C. J., Fu, L., Wang, H., Thomson, R. J., and Geiger, F. M.: Vibrational Mode Assignment of $\alpha$-Pinene by Isotope Editing: One Down, Seventy-One To Go, The Journal of Physical Chemistry A, 120, 2684–2690, https://doi.org/10.1021/acs.jpca.6b01995, pMID: 27063197, 2016.

Upshur, M. A., Vega, M. M., Bé, A. G., Chase, H. M., Zhang, Y., Tuladhar, A., Chase, Z. A., Fu, L., Ebben, C. J., Wang, Z., Martin, S. T., Geiger, F. M., and Thomson, R. J.: Synthesis and surface spectroscopy of $\alpha$-pinene isotopologues and their corresponding secondary organic material, Chem. Sci., 10, 8390–8398, https://doi.org/10.1039/C9SC02399B, 2019.

560